# Biogenically driven marine organic aerosol production over the West Pacific Ocean

Yujue Wang[1, 2, *], Yizhe Yi[1], Wei Xu[3, *], Yiwen Zhang[1], Shubin Li[1], Hong-Hai Zhang[4], Mingliang Gu[1],

Shibo Yan[5], Jialei Zhu[6], Chao Zhang[1, 2], Jinhui Shi[1, 2], Yang Gao[1, 2], Xiaohong Yao[1, 2], Huiwang Gao[1, 2]

[1] Frontiers Science Center for Deep Ocean Multispheres and Earth System, Key Laboratory of Marine Environment and Ecology, Ministry of Education of China, Ocean University of China, Qingdao, China
[2] Laboratory for Marine Ecology and Environmental Science, Qingdao Marine Science and Technology Center, Qingdao, China
[3] State Key Laboratory of Advanced Environmental Technology, Institute of Urban Environment, Chinese Academy of Sciences, Xiamen, China
[4] Key Laboratory of Marine Chemistry Theory and Technology, Ministry of Education, Ocean University of China, Qingdao 266100, China
[5] Third Institute of Oceanography, Ministry of Natural Resources, Siming District, Xiamen, Fujian 361005, China
[6] Institute of Surface-Earth System Science, School of Earth System Science, Tianjin University, Tianjin, China

* *Correspondence to*: Yujue Wang (wangyujue@ouc.edu.cn); Wei Xu (wxu@iue.ac.cn)

**Abstract.** Marine organic aerosols play crucial roles in cloud formation and climate regulation within the marine boundary layer. However, the abundance of marine primary organic carbon (MPOC) generated by sea spray and secondary organic carbon (MSOC) formed via gas-to-particle conversion or atmospheric oxidation/aging processes remains poorly quantified, which hinders our understanding on the climate effects of marine aerosols. In this work, two shipboard cruises were conducted over the West Pacific Ocean to estimate abundance and compositions of marine organic aerosols. We propose an observation-based approach to quantify the MPOC and MSOC using a combined parameterization of the observed $Na^+$ in fine aerosol particles and the surface chlorophyll-*a* (*Chl-a*), an indicator of marine biological activity. The parameterization approach of MPOC using $[Chl\text{-}a] \times [Na^+]^{0.45}$ was validated through comparing with the water-insoluble organic carbon in the aerosol samples. The estimated MPOC ($0.33 \pm 0.32$ μgC m$^{-3}$) averagely accounted for 56%−66% of the total organic carbon in the collected samples, which was mainly attributed to the protein-like substances transferred into the sea spray aerosols from seawater. Over the West Pacific Ocean, the MPOC and MSOC displayed peak concentrations over the regions 5°S–5°N ($0.64 \pm 0.56$ and $0.44 \pm 0.32$ μgC m$^{-3}$) and 35°N–40°N ($0.46 \pm 0.35$ and $0.51 \pm 0.30$ μgC m$^{-3}$). The variation and spatial distribution of MPOC and MSOC along the latitude were driven by the marine biological activities. High MSOC concentrations were also observed over the region of 15°N–20°N ($0.35 \pm 0.41$ μgC m$^{-3}$), which was due to an additional contribution by the oxidation of volatile organic precursors from the photochemical production of seawater organics. This study proposes a parameterization approach to quantify the MPOC and MSOC over the Pacific Ocean or other oceanic areas.

Our results highlight the marine biogenically driven formation of marine organic aerosols, and different quantitative relations
of MPOC with seawater *Chl-a* and other parameters are needed based on in-situ observations across oceanic regions.
**1 Introduction**
Marine aerosols are one of the most important natural aerosols on a global scale (De Leeuw et al., 2011; Quinn et al.,
2015b). Observation and modeling studies have proved that marine aerosols are an important source of cloud condensation
nuclei (CCN) and ice-nucleating particles (INPs) over remote oceanic areas, and play a vital role in Earth's radiation balance
(Demott et al., 2016; Quinn et al., 2017; Sinclair et al., 2020; Vergara-Temprado et al., 2017; Wolf et al., 2019; Xu et al.,
2022). Sea salt, sulfate, and organic matters (OM) make up the major components of marine aerosols, and the chemical
nature determines the hygroscopicity, ice nucleation, and climate impacts of marine aerosols (Huang et al., 2022; Zhao et al.,
2021). Marine organic aerosols (MOA) have attracted attention due to their effects on CCN formation over the remote ocean
(Zhao et al., 2021). Limited understanding on the formation, flux and composition of MOA results in the estimation
uncertainty of climate regulation by marine aerosols (Brooks and Thornton, 2018a; Quinn and Bates, 2011; Quinn et al.,
2015b).
Organics are a major fraction in marine aerosols, contributing 3%−90% of submicron aerosol mass (Huang et al., 2018;
O'dowd et al., 2004; O'dowd et al., 2008; Shank et al., 2012). MOA could be primarily released from the ocean surface or
secondarily formed via the oxidation and gas-to-particle conversion of volatile organic compounds (VOCs), including
dimethyl sulfide (DMS), isoprene, etc., in the marine boundary layer (Fu et al., 2011; Trueblood et al., 2019). Ocean surface
is one of the largest active reservoirs of organic carbon on Earth (~18%), resulting from phytoplankton, algal as well as the
related senescence and lysis (Hedges, 1992; Quinn and Bates, 2011). Wave breaking and bubble bursting at the ocean
surface would inject quantities of organic-enriched sea spray aerosols (SSA) into marine atmospheres (Hu et al., 2024;
Quinn et al., 2014). Organic matters are predominant in the fine or submicron SSA, which are usually dominated by water-
insoluble organic carbon (WIOC) (Cavalli, 2004b; Cravigan et al., 2020; Miyazaki et al., 2020). However, the majority of
the water-soluble organic carbon (WSOC) in MOA is contributed by secondary processes via the VOC oxidation or aged
organic aerosols (Schmitt-Kopplin et al., 2012; Trueblood et al., 2019).
A recent modeling study suggested that regional emission rates of MOA are largely related to the spatial distribution of
ocean biological productivity (Zhao et al., 2021). During phytoplankton blooms, the organic content elevated to as high as
63% of submicron aerosols, compared to a proportion of 15% during the low biological activity periods (O'dowd et al.,
2004). Seawater chlorophyll-a (*Chl-a*) or its combination with wind speed and aerosol size has been used to parameterize
the organic fraction in SSA (Gantt et al., 2012; Gantt et al., 2011). However, the abundance of various organics in SSA
remains highly uncertain and is a current challenge to understand their role in cloud formation (Albert et al., 2012; Brooks
and Thornton, 2018a).

Observation-based parameterization of primary and secondary MOA is urgently needed to constrain the modeling results (Brooks and Thornton, 2018b; Quinn et al., 2015a). In this work, two shipboard observations of atmospheric aerosols were conducted from the temperate to the tropical regions over the West Pacific Ocean (WPO) during spring and summer. Chemical compositions of marine aerosols, including organic carbon and inorganic ions, and seawater parameters were simultaneously obtained during the cruises. We derived a parameterization to estimate the primarily emitted organic aerosols in the SSA from wave breaking and bubble bursting, and separated the primary and secondary MOA based on the observation results. The derived formulation of primary MOA was validated by the measured water-insoluble organics and protein-like organic matter in marine aerosols, which are primarily generated by sea spray. We further investigated the spatial distribution, fluorescence characteristics of MOA, and the driving factors of MOA formation over the WPO. Our results provide an easy observation-based approach to divide the primary and secondary MOA based on the aerosol components and seawater *Chl-a*, as well as an observation-based parameterization of the primary MOA for further improving the parameterization of sea spray organic aerosols in large-scale models.

## 2 Materials and Methods

### 2.1 Cruises and sample collection

Two shipboard cruise observations were conducted over the West Pacific Ocean (Fig. 1). Cruise I was conducted in spring during 19 Feb.−9 April, 2022 on the R/V *KeXue* research vessel, and Cruise II was conducted in summer during 19 June−30 July, 2022 onboard of the R/V *Dongfanghong 3* research vessel. High-volume particle samplers (Qingdao Genstar Electronic Technology, China) were placed on the upper deck of the ship to collect the total suspended particles (TSP) and $PM_{2.5}$ (particles with a diameter of <2.5 µm) samples in marine atmospheres. To avoid the contamination of ship exhausts, the aerosol samplers were placed upwind on the foredeck of the ship. The quartz fiber filters were pre-baked at 500°C for 6 h before sample collection. The field blank aerosol sample was collected during each cruise.

Surface seawater samples were collected by a CTD (conductivity-temperature-depth) assembly (Seabird911). The concentration of the in-situ seawater *Chl-a* was measured using a fluorescence spectrophotometer (F-4700, Hitachi, Japan) (Wang et al., 2023a). Surface *Chl-a* concentrations were also obtained based on the satellite-derived data (Siemer et al., 2021; Tuchen et al., 2023). The satellite-derived *Chl-a* data were provided by Copernicus Marine Environmental Monitoring Service (CMEMS) with a spatial resolution of 4 km and a monthly temporal resolution (https://marine.copernicus.eu/). Here, we utilized the satellite-derived *Chl-a* data during March and June 2022 to support our conclusion. The concentration of soluble organic carbon in the seawater was measured by a total organic carbon (TOC) analyzer (TOC-L, Shimadzu, Japan). Air temperature and wind speed were monitored by the shipborne meteorological station. Surface net solar radiation (SSR) data were obtained from the hourly data of the ECMWF Reanalysis v5 (ERA5) product (Hersbach et al., 2020), with a spatial resolution of 0.25°. The 24-hr backward trajectories of air masses (Fig. S1) originating at 500 m above the ground level were calculated along the observation cruises every 24 hr using the HYSPLIT model (Version 5.2.1, NOAA).

## 2.2 Aerosol chemical composition analysis

An aliquot of the filter sample was extracted by Milli-Q water (>18.2 MΩ·cm) in ultrasonication, and filtered through 0.22 μm PTFE filters. The extracted solutions were analyzed by ion chromatograph systems (ICS-Aquion and ICS-2100 DIONEX) to obtain the concentrations of water-soluble inorganic ions ($Na^+$, $NH_4^+$, $K^+$, $Mg^{2+}$, $Ca^{2+}$, $Cl^-$, $NO_3^-$ and $SO_4^{2-}$) and methanesulfonic acid (MSA). The WSOC in the aerosol samples was measured by the TOC analyzer (TOC-L, Shimadzu, Japan). Organic carbon (OC) and elemental carbon (EC) were analyzed using a Sunset Laboratory thermal/optical carbon analyzer. Concentration of the water-insoluble organic carbon (WIOC) was calculated by the difference between OC and WSOC concentrations in each sample. The mass concentration of organic aerosols was calculated by multiplying OC by a conversion factor 1.6 (Wang et al., 2023b). The OM/OC conversion factor (1.6) was selected based on previous observation results of marine organic aerosols. Over the North Atlantic, an OM/OC mass ratio of 1.8 was adopted for WSOC based on the speciation of WSOC performed on the samples, and a conversion factor of 1.2 was applied for WIOC (Cavalli, 2004a). An average OM/OC ratio of 1.75 was observed in the submicron organic aerosol samples over the Atlantic Ocean (Huang et al., 2018). A higher proportion of water-soluble secondary organic aerosols (SOA), with higher OM/OC ratios than primary MOA, was observed in Huang et al. (2018) than in this study. Here, the proportions of WIOC are comparable to (summer cruise) or higher than (spring cruise) those of WSOC, and thus an OM/OC ratio of 1.6 was selected here. The mass concentrations of $PM_{2.5}$ or TSP were obtained by summing the measured OM, EC, and water-soluble ions in each aerosol sample. The aerosol samples with EC> 0.2 μgC m$^{-3}$ might be influenced by the ship exhausts (Lawler et al., 2020), which thus were excluded in our discussion. A total of 14 sets of aerosol samples during Cruise I and 17 sets of samples during Cruise II would be used for further discussion in this work.

## 2.3 Fluorescence spectra analysis

Filter aerosol samples were extracted by methanol and filtered through a 0.22 μm PTFE syringe filter. The methanol-extracted solutions were measured by a fluorescence spectrometer (F98, Lengguang Technology, China) to obtain the excitation (Ex) and emission (Em) spectra of MOA. Excitation−emission spectra were scanned within 200−600 nm using a 1 cm optical path length. Pre-processing of the fluorescence spectra data included instrument correction, inner filter correction, Raman and scattering removal, and blank subtraction, which was conducted according to Stedmon and Bro (2008) and Murphy et al. (2013). Fluorescent components in MOA were identified by excitation-emission matrix-parallel factor (EEM-PARAFAC) analysis (Murphy et al., 2013; Stedmon and Bro, 2008). The fluorescence intensity was reported using the unit of RU L$^{-1}$ m$^{-3}$ after considering the extracted solution volume and air volume of each sample (Fu et al., 2015).

## 3 Results and Discussion

### 3.1 Overview of marine organic aerosols during the cruises

The concentrations of the water-soluble ions and carbonaceous aerosols in the fine particles ($PM_{2.5}$) along the cruises are presented in Fig. 1. The average OC concentration in $PM_{2.5}$ was 0.67 µgC m$^{-3}$ (0.21−2.18 µgC m$^{-3}$) during the spring observation and 0.54 µgC m$^{-3}$ (0.12−1.42 µgC m$^{-3}$) during the summer observation. The EC concentrations were 0.066 $\pm$ 0.056 µgC m$^{-3}$ and 0.055 $\pm$ 0.052 µgC m$^{-3}$ during the spring and the summer observations, much lower than those observed over coastal areas typically influenced by continental outflows (Sahu et al., 2009; Zhang et al., 2025). The observed OC concentrations during our cruises were comparable to previous studies over the North Pacific Ocean (0.5−0.7 µgC m$^{-3}$), and lower than those observed at an island in the West Pacific Ocean (1.7$\pm$1.0 µgC m$^{-3}$) (Hoque et al., 2015; Hoque et al., 2017; Kunwar and Kawamura, 2014). Organic matters were the dominant components in the fine particles, which respectively contributed 18%−75% (40% on average) and 13%−74% (48% on average) of the $PM_{2.5}$ mass in spring and summer. This is consistent with previous findings that the organic fractions were dominant in the submicron marine aerosols (Facchini et al., 2008; O'dowd et al., 2004). Film drops could efficiently transfer hydrophobic organic compounds enriched in the air−water interface into the submicron aerosols, which explained the size-selective enrichment of organics in marine aerosols (Cochran et al., 2016b; Prather et al., 2013; Quinn et al., 2015a; Wang et al., 2017). The mass concentrations of $PM_{2.5}$ were calculated by summing the measured OM, EC, and water-soluble ions. Metal elements were not measured in this study, which contributed <3.5% of the marine aerosol mass concentration over the East China Sea (Hsu et al., 2010). Without considering the metal elements, we may overestimate the organic proportion in marine aerosols. During the sampling, positive artifacts of OC may exist due to the absorption of gaseous organic vapor on the filters, and negative artifacts may exist due to the evaporation of volatile organic compounds (Huebert and Charlson, 2000). The OC concentration was measured using thermal-optical analysis. Quantification uncertainty may be introduced due to the formation of pyrolyzed OC, which complicates the accurate determination of the OC/EC split point (Cao et al., 2025; Chow et al., 2004).

Taking the spring observation as an example, the OC mass in most $PM_{2.5}$ samples was roughly equal to that in the corresponding TSP samples (Fig. S2), which were simultaneously collected using two aerosol samplers during the cruise. The campaign-averaged OC concentrations were comparable in the $PM_{2.5}$ (0.67 µgC m$^{-3}$) and the TSP (0.69 µgC m$^{-3}$) samples. Thus, our further discussion on marine organic aerosols would focus on the results obtained from the $PM_{2.5}$ samples.

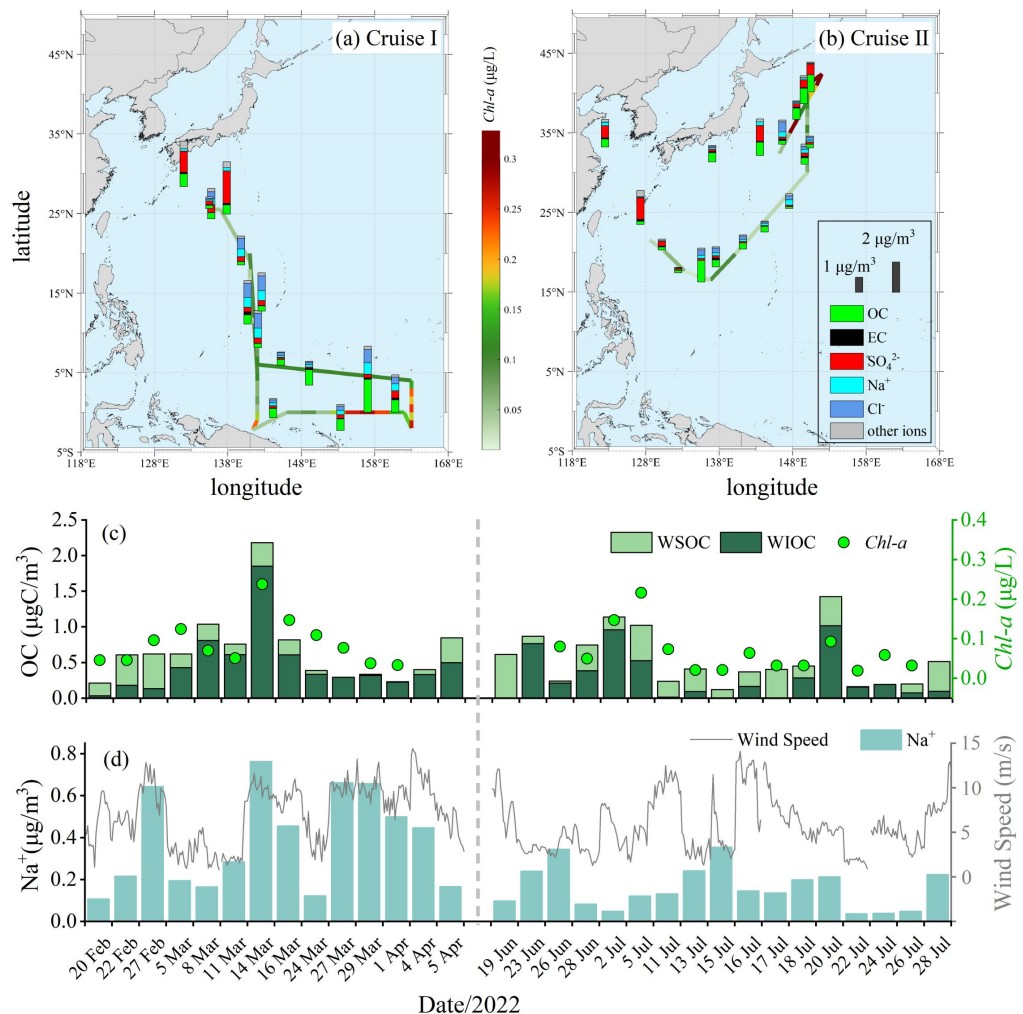

**Figure 1** Spatial distributions of water-soluble ions and carbonaceous aerosols (organic carbon, OC, and elemental carbon,
EC) in the $PM_{2.5}$ samples during (a) Cruise I conducted during spring, and (b) Cruise II conducted during summer. Time
series of (c) the water-soluble OC (WSOC), water-insoluble OC (WIOC), and $Chl\text{-}a$, and (d) $Na^+$ concentration in aerosol
samples and the wind speed during the cruises. In panels (a) and (b), the ship route is colored by the concentration of
seawater $Chl\text{-}a$. Other ions include $NO_3^-$, $NH_4^+$, $K^+$, $Mg^{2+}$ and $Ca^{2+}$.

The abundance of MOA displayed similar spatial distribution (Fig. 1) and strong or medium correlations with the sea
surface $Chl\text{-}a$ concentration (Cruise I: r = 0.81, p < 0.01; Cruise II: r = 0.67, p < 0.01), an indicator of the marine biological
activity (Brooks and Thornton, 2018a; Miyazaki et al., 2020). During the biologically active periods, the sea surface layer
was enriched in organics, which would be readily transferred into sea spray aerosols through wave breaking and bubble-
bursting processes (Cochran et al., 2016a; Cochran et al., 2017; Crocker et al., 2022; Wang et al., 2015). The correlation
coefficients between OC and EC were lower (Cruise I: r=0.48; Cruise II: r = 0.17) than those between OC and seawater $Chl\text{-}$
$a$, suggesting that the potential impacts of transported anthropogenic pollutants were limited during the cruises. The air

masses were mainly transported from open oceanic regions, and thus the impacts of terrestrial outflows were limited during the cruises (Fig. S1). The OC concentration levels in marine aerosols were higher during the spring cruise than during the summer cruise (Fig. 1c). This was due to the relatively higher phytoplankton activities along the cruise in spring, indicated by the higher seawater $Chl$-$a$ in spring ($0.09 \pm 0.06$ μg L$^{-1}$) than in summer ($0.07 \pm 0.05$ μg L$^{-1}$). However, the difference was not significant, with a $P$ value of 0.33. The highest OC concentration occurred on 14 March during the spring cruise, when the highest seawater $Chl$-$a$ ($0.24$ μg L$^{-1}$) was observed (Fig. 1). For the samples collected near the equator, the MOA or biogenic VOC precursors could also be transported from coastal oceanic regions of Papua New Guinea and Indonesia with higher marine biological activity and higher isoprene emission fluxes (Cui et al., 2023; Zhang and Gu, 2022). This could be an additional reason for the higher OC level during the spring cruise and the highest OC concentration observed on 14 March.

The observed concentrations of WSOC and WIOC over the open Pacific Ocean were lower than those observed in the atmosphere under severe influence of continental outflows (Sahu et al., 2009; Zhang et al., 2025). Marine organic aerosols were dominated by the water-insoluble fractions, with the WIOC/OC mass ratios of $70 \pm 27\%$ in spring and $48 \pm 35\%$ in summer (Fig. 1). The proportion of water-soluble organics in MOA over the WPO was lower than that observed over the East Asian marginal seas in autumn (75%), during which severe impacts of continental anthropogenic pollutants were observed (Zhang et al., 2025). The observed WIOC concentrations showed stronger correlations with the seawater $Chl$-$a$ ($r = 0.79$, $p < 0.01$ in spring and $r = 0.63$, $p < 0.05$ in summer) than the correlations between WSOC and $Chl$-$a$ ($r = 0.32$ in spring and $r = 0.42$ in summer). This indicated the closer linkage of marine biological-related organics with the WIOC than with the WSOC in marine aerosols. Marine phytoplankton could produce gel-like aggregates and contribute to extracellular polymer particles, water-insoluble polysaccharide-containing transparent exopolymer, and protein-containing organics, etc. in seawater (Aller et al., 2017; Lawler et al., 2020). These organic substances could be enriched in the surface seawater and then transferred into the atmospheric aerosols within the marine boundary layer. Previous studies suggested that seawater organics injected into aerosol particles through wave breaking or bubble bursting tend to be more hydrophobic and water insoluble (Cavalli, 2004b; Facchini et al., 2008; Miyazaki et al., 2010; O'dowd et al., 2004). Water-soluble organics in marine aerosols are usually related to the aged organic aerosols through long-range transportation or the SOA formed via the oxidation of marine reactive organic gases (Boreddy et al., 2018; De Jonge et al., 2024; Miyazaki et al., 2010). Reactive gaseous precursors of organic aerosols are widely observed over different oceanic regions (Tripathi et al., 2024; Tripathi et al., 2020; Wang et al., 2023a), which contribute to the SOA formation in the marine boundary layer.

## 3.2 Correlations of MOA with other parameters

The similar variation trends and good correlations between WIOC in marine aerosols and seawater $Chl$-$a$ (Fig. 1, 2) suggested the origins of MOA from seawater through ocean bubble bursting or wave breaking. Seawater $Chl$-$a$ is a widely used oceanic parameter to indicate the marine biological activity or the enrichment of organics in marine aerosols (O'dowd et al., 2004; O'dowd et al., 2008; Rinaldi et al., 2013; Spracklen et al., 2008), which has been employed to predict the organic fraction in marine aerosols. Over the West Pacific Ocean, we observed better correlations between OC or WIOC

concentrations and *Chl-a* than those between organic or water-insoluble organic mass fractions and *Chl-a* (Fig. 2a–2d). Some studies reported poor correlations between seawater *Chl-a* and the organic fraction in SSA, and proposed that the organic enrichment is also controlled by physical processes, especially the wind-driven sea spray production processes (De Leeuw et al., 2011; Lewis and Schwartz, 2004; Salter et al., 2014). Seawater *Chl-a* concentration is one of the most important factors driving the variation of organic fraction in the SSA, and they display good correlations when the wind speed does not vary a lot. However, wind speed should be combined with surface *Chl-a* to predict the organic fraction in SSA if the wind speed varies obviously during the observation or simulation periods (Gantt et al., 2011; Grythe et al., 2014). This is due to the influence of wind on the coverage of sea surface microlayer (SML) in the sea surface, which is enriched in organic compounds. For a given chemical composition of seawater, the largest coverage of sea surface by SML and a higher organic fraction in SSA are expected during calm winds. However, the SML would be destructed by mixing into the underlying seawater and the organic fraction in SSA decreased when surface wind exceeded 8 m s$^{-1}$ (Gantt et al., 2011). Thus, researchers usually combine wind speed with surface *Chl-a* to predict the organic fraction in SSA (Gantt et al., 2011; Grythe et al., 2014).

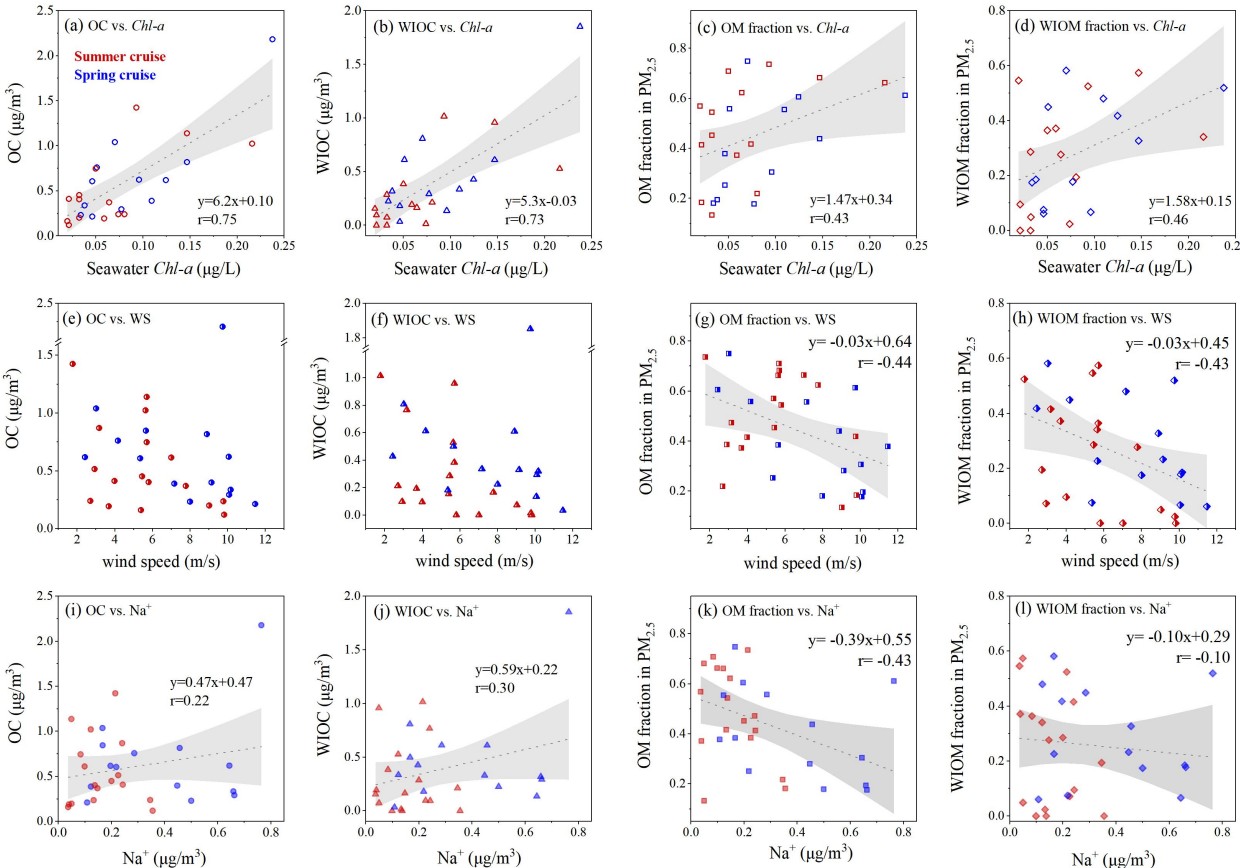

**Figure 2** The scatter plots of OC, WIOC concentrations or fractions in marine aerosols as a function of (a-d) surface seawater *Chl-a*, (e-h) wind speed (WS) and (i-l) [$Na^+$] in $PM_{2.5}$ samples during the two cruises. The data points during the springtime Cruise I and the summertime Cruise II are in blue and red, respectively. The regression line in each panel represents the correlation between the two parameters during the two cruises with a 95% confidence band.

During our cruises over the WPO, the concentrations of OC or WIOC in $PM_{2.5}$ showed a decreasing trend with the increase of wind speed (Fig. 2e, 2f). The organic fraction in marine aerosols displayed a negative correlation with the wind speed (Fig. 2g, 2h). The organic-enriched SML in the sea surface would be destructed under high wind speed conditions, which results in a decrease of organic substances transported into the SSA (Gantt et al., 2011). The concentration or proportion of $Na^+$ in the marine aerosols showed positive correlations with the wind speed during the observations (Fig. S3). Atmospheric SSA are primarily released as a mixture of inorganic sea salt and organic matters from the ocean surface. We observed weak positive correlations between OC or WIOC and $Na^+$ concentrations (Fig. 2i, 2j). We proposed that, for the filter-based observation or the samplings with a similar time resolution, $Na^+$ in fine particles could be used as a better indicator of the overall organic production levels than the wind speed in marine atmospheres. The [$Na^+$] represents the bulk sea salt abundance generated by wave breaking and bubble bursting, and reflects the overall effects of wind speeds and other meteorological conditions on SSA production during the period of filter sample collection. Russell et al. (2010) found strong correlations between ocean-derived submicron organic aerosols and $Na^+$ concentrations (Russell et al., 2010). It should be noted that dust storms also transport $Na^+$ to marine atmospheres, especially over the marginal seas (Zhang et al., 2025). When using $Na^+$ in marine aerosols as the indicator of SSA production, the $Na^+$ contributed by transported dust storms should be excluded, especially during dust episodes. For the collected TSP samples, the OC concentrations did not display an obvious correlation with the seawater *Chl-a* (Fig. S4). This is because the dominant production processes of OC and sea salts are different. Organic matters in marine aerosols are enriched in the submicron SSA, which is mainly formed by film drops from bursting bubble-cap films (Wang et al., 2017). In contrast, the majority of the sea salt mass exist in larger supermicron or coarse-mode particles generated by jet drops from the base of bursting bubbles (Wang et al., 2017).

### 3.3 Estimation of primary and secondary MOA

Based on the correlation analysis of the observed parameters, we proposed a parameterization scheme to separate the marine primarily-emitted OC (MPOC) in the SSA generated through wave breaking or bubble bursting processes and the secondarily formed organic carbon (MSOC) in the marine aerosols over WPO. For a given marine environment condition (a given *Chl-a*, wind speed, sea surface temperature (SST), etc.), the abundance of MPOC should be constant (Gantt et al., 2011). Seawater *Chl-a* concentration is the most important factors driving the variation of organic fraction in the SSA, and has been widely used to estimate the organic fraction in SSA (Gantt et al., 2011; Vignati et al., 2010). For given chemical composition of seawater, the largest organic fraction in SSA is expected during calm winds. An increase in wind speed above 3–4 m s$^{-1}$ will cause a rapid decrease of organic fraction due to the destructing of the SML coverage, and the lowest organic fraction is expected for wind exceeded 8 m s$^{-1}$ (Gantt et al., 2011). Seawater temperature is related to the production

efficiency and the number concentrations of SSA (Christiansen et al., 2019). In other words, it is a consistent relation between the MPOC and the sea surface *Chl-a* when the marine environment conditions remain stable. Sea surface *Chl-a* and wind speed have been utilized to parameterize the MPOC in global models (Gantt et al., 2012). Based on the shipboard observations in the present study, the mass ratio of the bulk OC (unit: µgC m$^{-3}$) versus seawater *Chl-a* (unit: µg L$^{-1}$) ranged from $3.0 \times 10^{-3}$ to $1.9 \times 10^{-2}$. The increased ratios or fitting line slope of OC versus *Chl-a* relevant to the lowest ones were attributed to the favorable marine conditions for the SSA generation, and the elevated contribution of SOA (e.g., methanesulfonic acid from DMS oxidation, isoprene SOA contributed by phytoplankton emission) (Barnes et al., 2006; De Jonge et al., 2024; Gupta et al., 2025; Ma et al., 2024; Wang et al., 2023b). The MSOC here includes the organic aerosols formed via gas-to-particle conversion of gaseous precursors and oxidation/aging processes of primary OC. The idea is conceptually similar to the classic OC/EC ratio method (Lim and Turpin, 2002; Turpin and Huntzicker, 1995), which uses EC as the tracer and has been widely used to estimate the primary and secondary OC in the continental atmospheres. Here we explored a formulation to estimate the MPOC and MSOC based on the observed OC and Na$^+$ in marine aerosols and the seawater *Chl-a*:

$$[OC] = [MPOC] + [MSOC] \qquad \text{Eq. 1}$$

$$[MPOC] = [Chl-a] \times \left( \frac{[OC]}{[Chl-a]} \right)_{SSA} \qquad \text{Eq. 2}$$

$$[MPOC] = ([Chl-a] \times [Na^+]^p) \times \left( \frac{[OC]}{[Chl-a] \times [Na^+]^p} \right)_{SSA} \qquad \text{Eq. 3}$$

where the [OC] is the total OC concentration in the marine aerosols, and [*Chl-a*] is the concentration of surface seawater *Chl-a*. The $\left( \frac{[OC]}{[Chl-a]} \right)_{SSA}$ in equation (2) represents the ratio of [OC] versus [*Chl-a*] in SSA, and $\left( \frac{[OC]}{[Chl-a] \times [Na^+]^p} \right)_{SSA}$ in equation (3) is [OC] versus ([*Chl-a*] × [Na$^+$]$^p$) in the primary SSA.

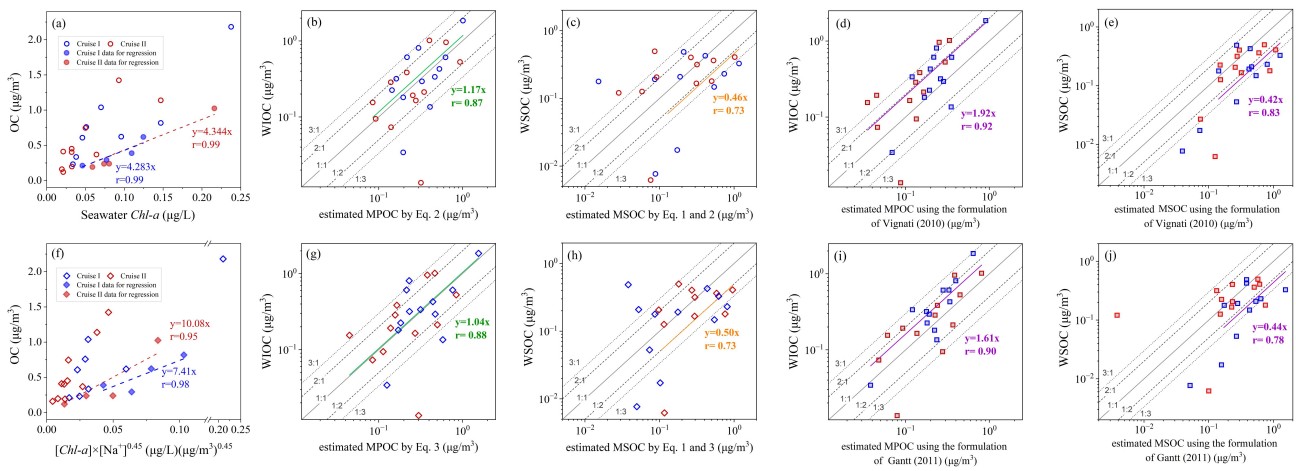

**Figure 3** The scatter plots of OC in marine aerosols as a function of (a) seawater [*Chl-a*] and (f) ([*Chl-a*] × [Na$^+$]$^{0.45}$) during the two cruises; (b, g) Comparison of WIOC and the estimated MPOC based on the regression in panel (a) and panel (f); (c,

h) Comparison of WSOC and the estimated MSOC; (d, i) Comparison of WIOC and the estimated MPOC, and (e, j)
Comparison of WSOC and the estimated MSOC using the formulation of Vignati (2010) and Gantt (2011). The dashed lines
in panels (a, f) are the regression line of [OC] and [$Chl$-$a$] or ([$Chl$-$a$] × [Na$^+$]$^{0.45}$) with 0–30% percentile ratios, indicated by
solid markers, during Cruise I (blue) and Cruise II (red). The regressions line in panels (b–e, g–j) represent the correlation
between WIOC and the estimated MPOC or between WSOC and the estimated MSOC in each panel during the two cruises.
Here, Eq. 2 or Eq. 3 is used to estimate the concentrations of MPOC based on the seawater [$Chl$-$a$] without or with
considering the simultaneously measured Na$^+$ concentrations as the input parameters. When $p$=0, Eq. 3 is the same
formulation as Eq. 2 without the Na$^+$ as an input parameter. Based on the correlation analysis, the MOA abundance was
mainly driven by the $Chl$-$a$ abundance. We used $Chl$-$a$ as the parameter to predict the concentration of MPOC in Eq. 2. For
the samples with the lowest 30% percentile of [OC]/[$Chl$-$a$] ratios, we proposed that the generation of organic aerosols was
dominated by the primary sea spray. The dataset with 0–30% percentile of [OC]/[$Chl$-$a$] ratios, indicated by the solid
markers in Fig. 3a, was used to calculate the fitting line of MPOC versus [$Chl$-$a$]. This is similar to the classic OC/EC ratio
method (Turpin and Huntzicker, 1995). The OC/EC ratios in POC is usually calculated based on the dataset with the lowest
10%−20% percentile OC/EC ratios observed during the campaign, which is then used to separate the POC and SOC in each
aerosol sample (Lim and Turpin, 2002; Yu et al., 2021). We used the data with the lowest 30% percentile of [OC]/[$Chl$-$a$]
ratios, considering the number of data points to calculate the fitting curve of MPOC versus [$Chl$-$a$]. With more data points,
the data with the lowest 10%−20% percentile [OC]/[$Chl$-$a$] ratios could be used to estimate the [MPOC]/[$Chl$-$a$] ratios, and
the estimated MPOC abundance may be a little higher than the results using the lowest 30% percentile data. The ratios of
[MPOC]/[$Chl$-$a$] were 4.28 during cruise I and 4.34 during cruise II (slopes of the fitting lines in Fig. 3a), which were then
used to estimate the MPOC during each cruise. The performance of MPOC parameterization was evaluated by comparing the
estimated concentrations of MPOC with the WIOC concentrations, which is generally considered as a proxy for MPOC (Fig.
3b). The average mass ratio of WIOC versus MPOC was 1.17 (r=0.87), and 69% of the data points fall within the 1:2 and 2:1
line (Fig. 3b). The shipboard observations suggested that the OC concentrations primarily generated by sea spray over the
WPO could be approximately estimated by 4.3×[$Chl$-$a$] when other related parameters were absent.
A combined parameterization scheme of multiplying seawater [$Chl$-$a$] by [Na$^+$]$^p$ was also used to predict the
concentration of MPOC (Eq. 3). A weak correlation between OC and Na$^+$ was observed here (Fig. 2i, 2j), and we thus
combined [Na$^+$] as the input parameter to reflect the variation of the bulk sea spray aerosol abundance. In the scatter plot of
OC and ([$Chl$-$a$] × [Na$^+$]$^p$), taking $p$=0.45 as an example in Fig. 3f, we proposed that the generation of organic matters were
dominated by the primary sea spray in the samples with the lowest 30% percentile of [OC]/([$Chl$-$a$] × [Na$^+$]$^p$) ratios. The
dataset with 0–30% percentile of [OC]/([$Chl$-$a$] ×[Na$^+$]$^{0.45}$) ratios, indicated by the solid markers in Fig. 3f, was used to
calculate the fitting line of MPOC versus [OC]/([$Chl$-$a$] ×[Na$^+$]$^{0.45}$). The fitting line was then employed to estimate the
MPOC in other marine aerosol samples based on the seawater $Chl$-$a$ and the aerosol Na$^+$ concentrations. In each sample, the
increased OC concentration relevant to the MPOC fitting line is attributed to the additional contribution by MSOC.
We compared the estimated MPOC and the measured WIOC to evaluate the performance of the MPOC
parameterization and determine the $p$ value in equation 3. Both the correlation coefficients (r) and the slopes of the fitting

line between WIOC and estimated MPOC are used to evaluate the performance of different MPOC parameterization approaches. It means that the estimated MPOC shows a similar variation trend to WIOC if with a r value closer to 1, and a good comparison with the WIOC mass concentrations if with a fitting line slope closer to 1. We tested the performance of the MPOC formulation when changing the $p$ value from 0−1, with an interval of 0.05. The variations of the fitting line slopes and correlation coefficients of WIOC and MPOC are shown in Fig. S3. When using a $p$ value of 0.35–0.65, the estimated MPOC matched well with WIOC concentrations, with the fitting line slopes of 1.03–1.05 and the r values of 0.86–0.88. When using p=0.45, both the fitting line slope (1.036) and r value (0.88) suggested an overall better performance than using other $p$ values (Fig. S5, 3g). Without the [$Na^+$] as an input parameter, the fitting line slope and r of WIOC and MPOC were respectively 1.17 and 0.87 (Fig. 3b), suggesting an underestimation of MPOC. In further analysis, we employed Eq. 3 with $p$=0.45 to estimate the MPOC. A total of 58% of the estimated data points fell within the WIOC/MPOC 2:1 and 1:2 lines (Fig. 3g), and 73% fell within the 3:1 and 1:3 lines, during the two cruises. The estimated MSOC matched better with the WSOC in the marine aerosols when using a combination of [$Chl$-$a$] and [$Na^+$] (equations 1 and 3) as the input parameters and considering the variation of sea spray aerosols (Fig. 3c, 3h). Based on equations 1 and 3, the estimated MSOC concentrations in half of the samples fall within the WSOC/MSOC 3:1 and 1:3 lines, and the fitting line slope (0.50) was closer to 1 (Fig. 3h). Using equations 1 and 2, the fitting line slope of WSOC and estimated MSOC was 0.46, and 46% of the estimated MSOC concentrations fall within the WSOC/MSOC 3:1 and 1:3 lines (Fig. 3c). It is noted that, based on the shipboard in-situ observation, we cannot exclude the potential impacts of gaseous precursors or aged organic aerosols long-range transported from terrestrial environments, which were mostly in the MSOC fraction. The organic aerosols transported from terrestrial environments were secondary or aged organic aerosols, and tend to be water-soluble organic compounds (Boreddy et al., 2018; De Jonge et al., 2024; Miyazaki et al., 2010). Based on the air mass back trajectories (Fig. S1) and the weak correlations between OC and EC stated in section 3.1, the impacts of transported continental outflows were limited during the cruises.

The MPOC was also estimated using the formulations in literatures (Gantt et al., 2011; Vignati et al., 2010) based on the observed seawater $Chl$-$a$, OC and $Na^+$ in aerosols as well as the wind speed observed during the cruises over the WPO (Fig. 3d, 3i). Vignati et al (2010) estimated the organic mass fraction in sea spray aerosol ($OM_{SSA}$) using seawater [$Chl$-$a$]: $\%OM_{SSA} = 43.5 \times [Chl$-$a](mg\ m^{-3}) + 13.805$. Gantt et al (2011) predicted the $OM_{SSA}$ using a combination of [$Chl$-$a$] and 10 m wind speed ($U_{10}$): $OM_{SSA}(\text{Chl} - \text{a}, U_{10}) = \frac{OM_{SSA}^{max}}{1+\exp{(-2.63[Chl-a]+0.18U_{10})}}$ , where $OM_{SSA}^{max}$ is the maximum $OM_{SSA}$ observed during the cruises. The estimated MPOC displayed good correlations with the observed WIOC. However, the abundance of MPOC was underestimated approximately by 38%−48% through comparing with the WIOC concentrations (Fig. 3d, 3i). The estimated MSOC using the parameterizations from Gantt et al. (2011) or Vignati et al. (2010) showed similar variation trends to the WSOC in the collected aerosols samples. The comparison of the estimated MSOC and the WSOC concentrations using formulations in literatures (slopes in Fig. 3e, 3j), however, were not as good as those estimated in this study (slopes in Fig. 3h). The MPOC source functions in Gantt et al. (2011) and Vignati et al. (2010) were proposed based

on the observation over the North Atlantic, which has been widely employed in large-scale models. These parameterizations perform well to trace the variation trends of MPOC. However, they might lead to an underestimation of the primary MOA over the West Pacific Ocean. This is mainly due to different seawater compositions, marine environment or atmospheric meteorological conditions in the North Atlantic and the West Pacific Oceans, which result in different quantitative relations between seawater *Chl-a* and MPOC in these oceanic regions. What's more, the seawater *Chl-a* was determined using the spatial average of the satellite-derived *Chl-a* concentrations in Gantt et al. (2011). This could be an additional reason for the different parameterizations between *Chl-a* and MPOC compared with the results based on the in-suit measured *Chl-a* in this work. The results highlight different quantitative relations of MPOC with seawater *Chl-a* and other parameters in different areas, which are needed to be provided through in-situ observations across different oceanic regions and to constrain in global models.

### 3.4 Spatial distribution and driving factors of primary and secondary MOA

Based on the validated formulation, concentrations of MPOC and MSOC in the marine aerosols over the WPO are estimated. Here we employed Eq. 3, with $p$=0.45, for the estimation of MPOC. The concentrations and relative contributions of MPOC and MSOC along the latitude are shown in Fig. 4. The estimated MPOC was respectively 0.43 ± 0.40 and 0.24 ± 0.21 µgC m$^{-3}$, averagely accounting for 66% ± 27% and 56% ± 30% of the total OC in marine aerosols, during the springtime Cruise I and the summertime Cruise II. The dominant contribution of MOA by the marine fresh carbon pool was also observed during the Arctic cruises, during which the MPOC contributed 80% of the carbonaceous fraction based on the stable carbon isotopic signature (Gu et al., 2023). The estimated MSOC concentrations were comparable in spring (0.25 ± 0.28 µgC m$^{-3}$) and in summer (0.27 ± 0.30 µgC m$^{-3}$) over the WPO. The SOA fraction among the total organic aerosols was higher during the summer cruise (44% on average) than during the spring (34% on average).

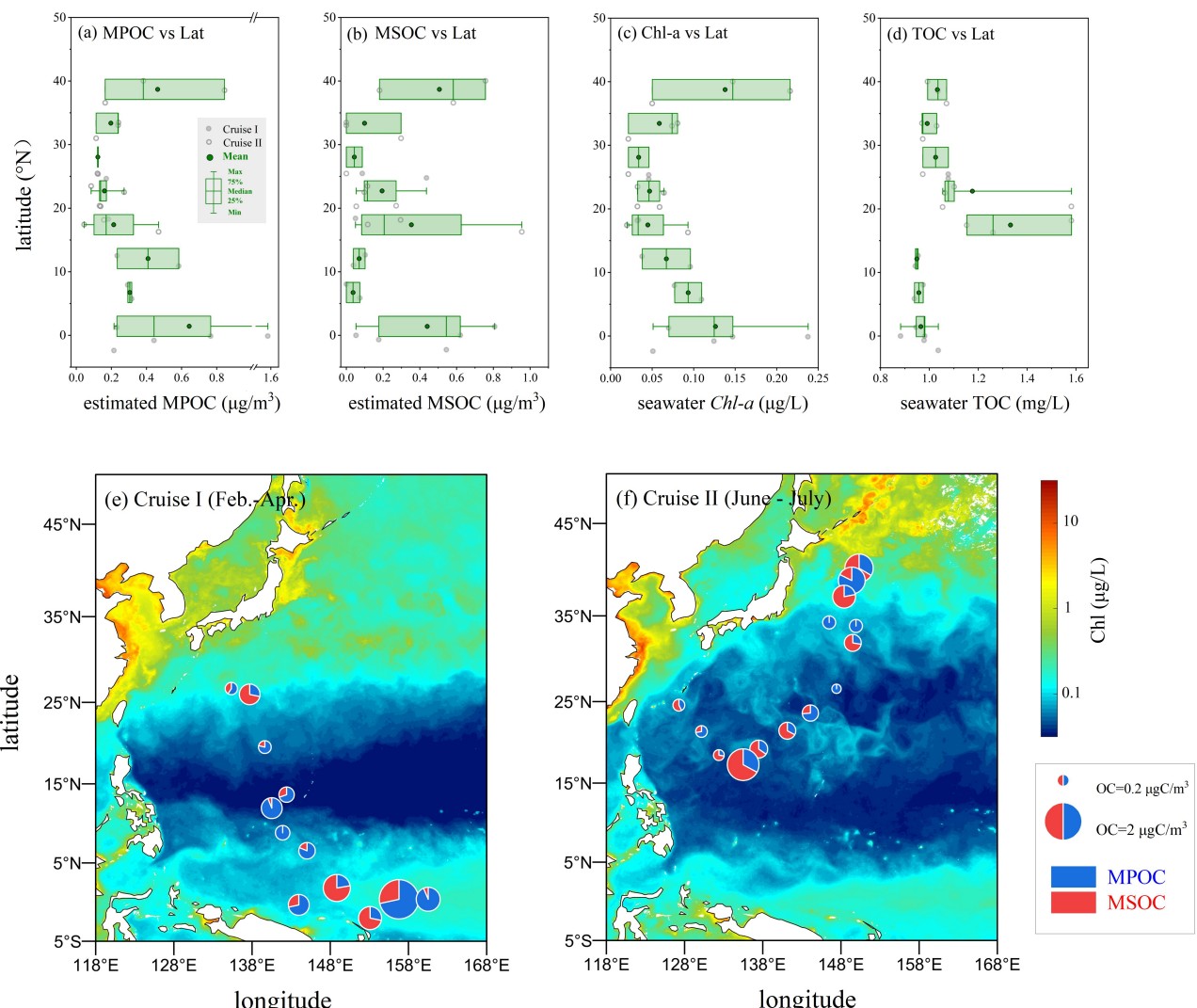

356

**Figure 4** The variations of (a) estimated MPOC, (b) MSOC, (c) seawater *Chl-a* and (d) TOC along the latitude during the two cruises over the WPO. Spatial distributions of the estimated MPOC and MSOC during (e) the springtime Cruise I, and (f) the summertime Cruise II. Ocean is coloured by the sea surface *Chl-a* concentrations in March (panel e) and July (panel f), 2022. The marker size in panels (e, f) represents the observed OC concentration in each sample.

Both MPOC and MSOC displayed high concentrations over the oceanic regions among 5°S–5°N (0.64 ± 0.56 and 0.44 ± 0.32 µgC m$^{-3}$) and 35°N–40°N (0.46 ± 0.35 and 0.51 ± 0.30 µgC m$^{-3}$), which were consistent to the spatial distribution of the sea surface *Chl-a* (Fig. 4). In contrast to the findings over the North Atlantic that plankton had little impact on the chemical compositions of SSA (e.g., organic mass fraction) (Bates et al., 2020), we observed a positive correlation between MOA and seawater *Chl-a* and the driving effects of surface *Chl-a* on the abundance of primary MOA over the WPO (Fig. 2,

4). In addition, the observation areas within 35°N–40°N were the Kuroshio Oyashio Extension (KOE) region, where the nutrient enrichment driven by upwelling favored the phytoplankton growth and resulted in elevated seawater *Chl-a* levels (Wang et al., 2023a). The extreme physical disturbance in the KOE further promoted the sea spray-generated organics from seawater as well as the production of VOCs from phytoplankton. Based on the air mass back trajectories (Fig. S1), the impacts of transported terrestrial outflows were limited among the observation regions. Marine organic aerosols or biogenic VOC precursors could also be transported from coastal oceanic regions with higher *Chl-a* levels and higher isoprene emission fluxes (Cui et al., 2023; Zhang and Gu, 2022), which could be an additional reason for the higher MOA concentrations within 5°S–5°N and 35°N–40°N.

The MSOC also displayed a peak over the areas among 15°N–20°N (0.35 ± 0.41 µgC m$^{-3}$, Fig. 4b), which could be attributed to the additional contribution by abiotic VOC precursors from the photochemical production in the sea surface microlayer and their further oxidation in marine boundary layer (Bruggemann et al., 2018). Previous studies suggested that interfacial photochemical degradation of dissolved organic matters in seawater could be an important source of marine VOCs (e.g., isoprene) on a global scale (Bruggemann et al., 2018; Cui et al., 2023; Wang et al., 2023a; Yu and Li, 2021). For remote oceanic regions with high solar radiation but low biological activities, interfacial photochemistry of surface organics could be a major source of abiotic VOCs in the marine boundary layer (Bruggemann et al., 2018; Cui et al., 2023). Higher concentration levels of the surface seawater TOC concentrations were observed along the summer cruise within 15°N–20°N (Fig. 4d). The strong solar radiation during the summertime (19 June- 30 July) Cruise II, as shown in Fig. S6, favored the photochemical VOC production and the SOA formation in marine atmospheres. During the summer cruise, the estimated MSOC/OC ratios over the oceanic regions of 15°N–20°N were 65%−72%, and the SOA formation drove the elevation of MOA concentrations over this area during summer (Fig. 4f).

Aerosol samples were collected among 15°N−30°N during both the spring and the summer cruises, which were compared to elaborate the seasonal difference. The variations of the estimated MPOC and MSOC along the latitude are shown in Fig. S7, S8. Among the observation region within 15°N−30°N, the average MPOC was comparable in spring (0.16 µgC m$^{-3}$) and summer (0.18 µgC m$^{-3}$), with the average *Chl-a* concentration 0.042 and 0.044 µg L$^{-1}$, respectively. Among the oceanic regions with similar concentrations of seawater *Chl-a*, the MPOC abundance in marine aerosols was comparable without seasonal difference. Among 15°N−30°N, the elevation of MPOC concentrations was consistent with the elevated seawater *Chl-a* concentration without seasonal difference (Fig. S7a). This is consistent with the finding that marine biogenic activities drive the MPOC production. The average MSOC concentration was 0.24 µgC m$^{-3}$ within 15°N−30°N in summer, higher than that in spring (0.19 µgC m$^{-3}$). The elevated MSOC was driven by the increase of seawater TOC concentrations (Fig. S8b). What's more, the stronger solar radiation in summer (Fig. S6) favored the photochemical VOC production in SML, their further photo-oxidation reactions, and the MSOC formation in the atmosphere.

## 3.5 Fluorescence characteristics of MOA

The fluorescence spectrum of MOA was analyzed to gain a further understanding on the composition characteristics of MOA over the WPO. Based on the EEM PARAFAC analysis, three fluorescent components were identified in marine organic aerosols during each cruise observation (Fig. 5, S4). Similar fluorescence components were resolved during the Cruise I (Fig. 5a-5c) and Cruise II (Fig. 5d-5f). Each component was named based on the fluorescence characteristics and the temporal variation of the fluorescent intensity. Component 1 (C1) shows a peak (Ex/Em = 285/307 nm) identical to the protein-like substances (PRLIS) (Chen et al., 2016b). The PRLIS are enriched in the surface seawater and could be injected into SSA via bubble bursting (Miyazaki et al., 2018a; Santander et al., 2021). The similar variations of C1 intensity and $Na^+$ in the marine aerosols, especially during the summer observation (Fig. S9), suggested the origins of PRLIS from marine biological materials (Fu et al., 2015; Santander et al., 2022). Thus, C1 was designated as marine PRLIS. Component 2 (C2) has a peak Ex/Em = 320-335/389 nm (Fig. 5), which is related to terrestrial humic-like substances (HULIS) (Chen et al., 2016b). Component 3 (C3) displayed the fluorescence characteristics of oxygenated HULIS, with a peak Ex/Em = 365-370/450-455 nm (Fig. 5). The intensities of C2 and C3 showed similar variations to the concentrations of EC and sulfate in marine aerosols (Fig. S9), which indicated their sources related to combustion emission and secondary formation (Tang et al., 2024). Oxygenated HULIS included the secondarily-formed and aged organic aerosols from both terrestrial and marine sources.

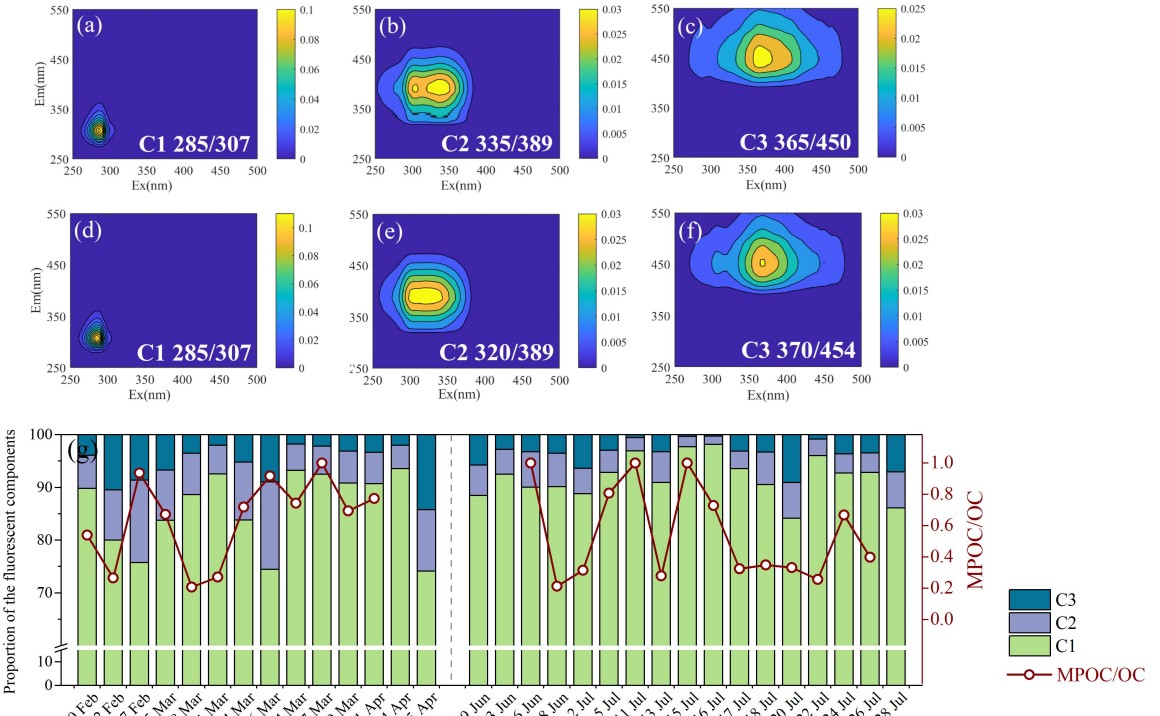

**Figure 5** (a-f) The excitation (Ex) and emission (Em) spectra of the identified fluorescent components: C1: marine protein-like substances (PRLIS), C2: terrestrial humic-like substances (HULIS), C3: oxygenated HULIS. (g) Time series of the fluorescent component relative abundances and the MPOC/OC mass ratios in the marine aerosols over the West Pacific Ocean.

The fluorescent components of the MOA were dominated by the PRLIS primarily emitted by the sea spray (C1 shown in Fig. 5a, 5d), which contributed 74%−94% (86% on average) and 84%−98% (92% on average) of the MOA fluorescent intensity during Cruise I and Cruise II, respectively (Fig. 5g). It is noted that the proportion of different fluorescent compounds did not represent their mass contributions, as the florescent efficiency of organic compounds was related to their chemical structures. Organic molecules with substantial conjugation of $\pi$-bonds or double bound structures are known to be especially efficient at emitting fluorescence, particularly when N atoms are present (Chen et al., 2016a; Pöhlker et al., 2012). Amino acids, vitamins, and humic-like substances have been identified as efficient fluorophores (Graber and Rudich, 2006; Laskin et al., 2015; Pöhlker et al., 2012). The sea-to-air transfer of phytoplankton-produced protein-containing organics leads to a significant enhancement of fluorescent compounds in SSA (Aller et al., 2017; Lawler et al., 2020; Miyazaki et al., 2018b). The PRLIS, or named protein-like organic matter (PLOM), has been identified as a common component in the oceanic organic matter, and enriched in marine aerosols (Chen et al., 2016b). However, the biogenic SOA (e.g., isoprene oxidation products abundant in marine atmospheres) molecules, without conjugated double bounds, are weakly fluorescent or do not display fluorescent properties (Carlton et al., 2009; Laskin et al., 2015). Thus, the WSOC contributed by biogenic SOA was not included in the detected fluorescent components, and the observed proportions of PRLIS emitted by sea sprays was higher than those of the WIOC mass contribution in the marine aerosols.

During the summertime Cruise II, the fluorescent intensity and the relative contribution of marine-emitted PRLIS (C1) were higher than those during Cruise I. During the summer cruise, the contribution of the marine PRLIS among the total fluorescent organic aerosols displayed a similar variation trend to the mass fraction of the estimated MPOC (Fig. 5g). The variation of the marine PRLIS (C1) intensity was consistent to the seawater *Chl-a* concentration in summer (Fig. S9). This further indicated the dominant contribution of primary MOA in marine organic aerosols, which could be attributed to the marine biological materials and injected into the atmosphere through bubble bursting. The marine biological PRLIS could be related to tryptophan-like or tyrosine-like components as well as the non-nitrogen-containing organic compounds in atmospheric aerosols (Chen et al., 2016b).

## 4 Summary

In-situ shipboard observations were conducted to investigate the abundance and composition of MOA over the open Pacific Ocean. We proposed a formulation to separate and estimate the primary and secondary MOA based on the seawater *Chl-a* or its combination with $Na^+$ in marine aerosols. Based on the validated formulation, the estimated MPOC accounted for 56%−66% of the total OC in the marine aerosol samples, which were mostly related to the protein-like substances from seawater biological materials. Both the MPOC and the MSOC displayed peak concentrations among 5°S–5°N and 35°N–

40°N over the West Pacific Ocean. The spatial distribution of MOA along the latitude was driven by the marine biological activities, indicated by the seawater *Chl-a*. For the secondary MOA, high concentrations were also observed over the region of 15°N–20°N, which was attributed to an additional contribution by the secondary oxidation of VOCs generated from the photochemical production of seawater organics.

This study provides a parameterization to estimate the primary and secondary MOA based on the shipboard observation evidence, and highlights the marine biogenically driven MOA formation over the North Pacific Ocean. For the observation studies, our results provide an easy approach to separate the primary and secondary MOA with different chemical natures, based on the seawater *Chl-a* and aerosol components (OC, $Na^+$). The approach is not dependent on the organic tracers, usually obtained through complex analysis procedures, or limited to the time resolution of sample collection. In previous studies, fractions of organics in marine aerosols have been estimated based on an empirical relationship of satellite-derived oceanic *Chl-a*, or a combination with wind speed and aerosol size distribution (Gantt et al., 2012; Li et al., 2024; Wang et al., 2024). Here, we gain the quantitative relations of primarily-generated marine organic aerosols with sea salts and *Chl-a* based on measurement results of the marine aerosols and seawater. For the modelling studies, the sea salt flux has been better estimated than that of marine organic aerosols in global models (Gantt and Meskhidze, 2013). The MPOC formulation here would help to improve the parameterization of MOA in models and better understand the climate effects of marine aerosols on a global scale.

**Data availability**

The data is available via https://zenodo.org/records/16831992 (Wang, 2025)

**Author contributions**

Y.W. designed the research. Y.Y., Y.Z., S.L., H.Z., and S.Y. conducted the measurements. Y.Y. and Y.W analyzed the data. Y. W., Y.Y. and W.X. wrote the manuscript with contributions from all authors.

**Acknowledgments**

This study was supported by the National Key Research and Development Program of China (2024YFC2815800, 2023YFC3705503), the National Natural Science Foundation of China (42205103; 42411540229), the Taishan Scholars of Shandong Province, China (tsqn202306101), the Fundamental Research Funds for the Central Universities (202441011), and

477 the Shandong Provincial Natural Science Foundation (ZR2022QD105). The Fund of Key Laboratory of Marine Ecological
478 Conservation and Restoration, Ministry of Natural Resources/ Fujian Provincial Key Laboratory of Marine Ecological
479 Conservation and Restoration (EPR2025009); State Environmental Protection Key Laboratory of Formation and Prevention
480 of Urban Air Pollution Complex (No. 2025080172).

481 Date and samples were collected onboard of R/V *Dongfanghong 3* and R/V *KeXue* implementing the open research
482 cruise NORC2024-584 supported by NSFC Shiptime Sharing Project (42349584), the Laoshan Laboratory (LSKJ202201701,
483 LSKJ202400202), and the Fundamental Research Funds for the Central Universities (202372001, 202472001)

484 **Competing interests**

485 The authors declare no conflict of interests.

486

487 **References**

488 Albert, M. F. M. A., Schaap, M., Manders, A. M. M., Scannell, C., O'Dowd, C. D., and de Leeuw, G.: Uncertainties in the
489 determination of global sub-micron marine organic matter emissions, Atmospheric Environment, 57, 289-300,
490 10.1016/j.atmosenv.2012.04.009, 2012.
491 Aller, J. Y., Radway, J. C., Kilthau, W. P., Bothe, D. W., Wilson, T. W., Vaillancourt, R. D., Quinn, P. K., Coffman, D. J.,
492 Murray, B. J., and Knopf, D. A.: Size-resolved characterization of the polysaccharidic and proteinaceous components of sea
493 spray aerosol, Atmospheric Environment, 154, 331-347, 10.1016/j.atmosenv.2017.01.053, 2017.
494 Barnes, I., Hjorth, J., and Mihalopoulos, N.: Dimethyl sulfide and dimethyl sulfoxide and their oxidation in the atmosphere,
495 Chem. Rev., 106, 940-975, 10.1021/cr020529+, 2006.
496 Bates, T. S., Quinn, P. K., Coffman, D. J., Johnson, J. E., Upchurch, L., Saliba, G., Lewis, S., Graff, J., Russell, L. M., and
497 Behrenfeld, M. J.: Variability in Marine Plankton Ecosystems Are Not Observed in Freshly Emitted Sea Spray Aerosol Over
498 the North Atlantic Ocean, Geophys. Res. Lett., 47, 10.1029/2019gl085938, 2020.
499 Boreddy, S. K. R., Haque, M. M., and Kawamura, K.: Long-term (2001–2012) trends of carbonaceous aerosols from a
500 remote island in the western North Pacific: an outflow region of Asian pollutants, Atmospheric Chemistry and Physics, 18,
501 1291-1306, 10.5194/acp-18-1291-2018, 2018.
502 Brooks, S. D. and Thornton, D. C. O.: Marine Aerosols and Clouds, Annual Review of Marine Science, 10, 289-313,
503 10.1146/annurev-marine-121916-063148, 2018a.
504 Brooks, S. D. and Thornton, D. C. O.: Marine Aerosols and Clouds, Ann Rev Mar Sci, 10, 289-313, 10.1146/annurev-
505 marine-121916-063148, 2018b.
506 Bruggemann, M., Hayeck, N., and George, C.: Interfacial photochemistry at the ocean surface is a global source of organic
507 vapors and aerosols, Nat Commun, 9, 2101, 10.1038/s41467-018-04528-7, 2018.
508 Cao, C., Yu, X., Marco Wong, W. H., Sun, N., Zhang, K., Sun, Z., Chen, L., Wu, C., Wang, G., and Yu, J. Z.: An
509 Instrumental Method for the Simultaneous Determination of Organic Carbon, Elemental Carbon, Inorganic Nitrogen, and
510 Organic Nitrogen in Aerosol Samples, J. Geophys. Res., [Atmos.], 130, 10.1029/2025jd043904, 2025.
511 Carlton, A. G., Wiedinmyer, C., and Kroll, J. H.: A review of Secondary Organic Aerosol (SOA) formation from isoprene,
512 Atmos. Chem. Phys., 9, 4987-5005, 10.5194/acp-9-4987-2009, 2009.
513 Cavalli, F.: Advances in characterization of size-resolved organic matter in marine aerosol over the North Atlantic, J.
514 Geophys. Res., 109, D24215, 10.1029/2004jd005137, 2004a.
515 Cavalli, F.: Advances in characterization of size-resolved organic matter in marine aerosol over the North Atlantic, Journal
516 of Geophysical Research, 109, 10.1029/2004jd005137, 2004b.
517 Chen, Q., Ikemori, F., and Mochida, M.: Light Absorption and Excitation-Emission Fluorescence of Urban Organic Aerosol
518 Components and Their Relationship to Chemical Structure, Environ. Sci. Technol., 50, 10859-10868,
519 10.1021/acs.est.6b02541, 2016a.

Chen, Q., Miyazaki, Y., Kawamura, K., Matsumoto, K., Coburn, S., Volkamer, R., Iwamoto, Y., Kagami, S., Deng, Y., Ogawa, S., Ramasamy, S., Kato, S., Ida, A., Kajii, Y., and Mochida, M.: Characterization of Chromophoric Water-Soluble Organic Matter in Urban, Forest, and Marine Aerosols by HR-ToF-AMS Analysis and Excitation-Emission Matrix Spectroscopy, Environmental Science & Technology, 50, 10351-10360, 10.1021/acs.est.6b01643, 2016b.

Chow, J. C., Watson, J. G., Chen, L. W., Arnott, W. P., Moosmuller, H., and Fung, K.: Equivalence of elemental carbon by thermal/optical reflectance and transmittance with different temperature protocols, Environ. Sci. Technol., 38, 4414-4422, 10.1021/es034936u, 2004.

Christiansen, S., Salter, M. E., Gorokhova, E., Nguyen, Q. T., and Bilde, M.: Sea spray aerosol formation: Laboratory results on the role of air entrainment, water temperature, and phytoplankton biomass, Environ. Sci. Technol., 53, 13107-13116, 10.1021/acs.est.9b04078, 2019.

Cochran, R. E., Laskina, O., Jayarathne, T., Laskin, A., Laskin, J., Lin, P., Sultana, C., Lee, C., Moore, K. A., Cappa, C. D., Bertram, T. H., Prather, K. A., Grassian, V. H., and Stone, E. A.: Analysis of Organic Anionic Surfactants in Fine and Coarse Fractions of Freshly Emitted Sea Spray Aerosol, Environmental Science & Technology, 50, 2477-2486, 10.1021/acs.est.5b04053, 2016a.

Cochran, R. E., Laskina, O., Jayarathne, T., Laskin, A., Laskin, J., Lin, P., Sultana, C., Lee, C., Moore, K. A., Cappa, C. D., Bertram, T. H., Prather, K. A., Grassian, V. H., and Stone, E. A.: Analysis of Organic Anionic Surfactants in Fine and Coarse Fractions of Freshly Emitted Sea Spray Aerosol, Environ. Sci. Technol., 50, 2477-2486, 10.1021/acs.est.5b04053, 2016b.

Cochran, R. E., Laskina, O., Trueblood, J. V., Estillore, A. D., Morris, H. S., Jayarathne, T., Sultana, C. M., Lee, C., Lin, P., Laskin, J., Laskin, A., Dowling, J. A., Qin, Z., Cappa, C. D., Bertram, T. H., Tivanski, A. V., Stone, E. A., Prather, K. A., and Grassian, V. H.: Molecular Diversity of Sea Spray Aerosol Particles: Impact of Ocean Biology on Particle Composition and Hygroscopicity, Chem, 2, 655-667, 10.1016/j.chempr.2017.03.007, 2017.

Cravigan, L. T., Mallet, M. D., Vaattovaara, P., Harvey, M. J., Law, C. S., Modini, R. L., Russell, L. M., Stelcer, E., Cohen, D. D., Olsen, G., Safi, K., Burrell, T. J., and Ristovski, Z.: Sea spray aerosol organic enrichment, water uptake and surface tension effects, Atmospheric Chemistry and Physics, 20, 7955-7977, 10.5194/acp-20-7955-2020, 2020.

Crocker, D. R., Kaluarachchi, C. P., Cao, R., Dinasquet, J., Franklin, E. B., Morris, C. K., Amiri, S., Petras, D., Nguyen, T., Torres, R. R., Martz, T. R., Malfatti, F., Goldstein, A. H., Tivanski, A. V., Prather, K. A., and Thiemens, M. H.: Isotopic Insights into Organic Composition Differences between Supermicron and Submicron Sea Spray Aerosol, Environmental Science & Technology, 56, 9947-9958, 10.1021/acs.est.2c02154, 2022.

Cui, L., Xiao, Y., Hu, W., Song, L., Wang, Y., Zhang, C., Fu, P., and Zhu, J.: Enhanced dataset of global marine isoprene emissions from biogenic and photochemical processes for the period 2001–2020, Earth System Science Data, 15, 5403-5425, 10.5194/essd-15-5403-2023, 2023.

de Jonge, R. W., Xavier, C., Olenius, T., Elm, J., Svenhag, C., Hyttinen, N., Nieradzik, L., Sarnela, N., Kristensson, A., Petaja, T., Ehn, M., and Roldin, P.: Natural Marine Precursors Boost Continental New Particle Formation and Production of Cloud Condensation Nuclei, Environmental Science & Technology, 58, 10956-10968, 10.1021/acs.est.4c01891, 2024.

de Leeuw, G., Andreas, E. L., Anguelova, M. D., Fairall, C. W., Lewis, E. R., O'Dowd, C., Schulz, M., and Schwartz, S. E.: Production flux of sea spray aerosol, Rev. Geophys., 49, 10.1029/2010rg000349, 2011.

DeMott, P. J., Hill, T. C., McCluskey, C. S., Prather, K. A., Collins, D. B., Sullivan, R. C., Ruppel, M. J., Mason, R. H., Irish, V. E., Lee, T., Hwang, C. Y., Rhee, T. S., Snider, J. R., McMeeking, G. R., Dhaniyala, S., Lewis, E. R., Wentzell, J. J., Abbatt, J., Lee, C., Sultana, C. M., Ault, A. P., Axson, J. L., Diaz Martinez, M., Venero, I., Santos-Figueroa, G., Stokes, M. D., Deane, G. B., Mayol-Bracero, O. L., Grassian, V. H., Bertram, T. H., Bertram, A. K., Moffett, B. F., and Franc, G. D.: Sea spray aerosol as a unique source of ice nucleating particles, The Proceedings of the National Academy of Sciences, 113, 5797-5803, 10.1073/pnas.1514034112, 2016.

Facchini, M. C., Rinaldi, M., Decesari, S., Carbone, C., Finessi, E., Mircea, M., Fuzzi, S., Ceburnis, D., Flanagan, R., Nilsson, E. D., de Leeuw, G., Martino, M., Woeltjen, J., and O'Dowd, C. D.: Primary submicron marine aerosol dominated by insoluble organic colloids and aggregates, Geophys. Res. Lett., 35, 10.1029/2008gl034210, 2008.

Fu, P., Kawamura, K., and Miura, K.: Molecular characterization of marine organic aerosols collected during a round-the-world cruise, Journal of Geophysical Research, 116, 10.1029/2011jd015604, 2011.

Fu, P., Kawamura, K., Chen, J., Qin, M., Ren, L., Sun, Y., Wang, Z., Barrie, L. A., Tachibana, E., Ding, A., and Yamashita, Y.: Fluorescent water-soluble organic aerosols in the High Arctic atmosphere, Scientific Reports, 5, 9845, 10.1038/srep09845, 2015.

Gantt, B. and Meskhidze, N.: The physical and chemical characteristics of marine primary organic aerosol: a review, Atmospheric Chemistry and Physics, 13, 3979-3996, 10.5194/acp-13-3979-2013, 2013.

Gantt, B., Meskhidze, N., Facchini, M. C., Rinaldi, M., Ceburnis, D., and O'Dowd, C. D.: Wind speed dependent size-resolved parameterization for the organic mass fraction of sea spray aerosol, Atmos. Chem. Phys., 11, 8777-8790, 10.5194/acp-11-8777-2011, 2011.

Gantt, B., Johnson, M. S., Meskhidze, N., Sciare, J., Ovadnevaite, J., Ceburnis, D., and O'Dowd, C. D.: Model evaluation of marine primary organic aerosol emission schemes, Atmos. Chem. Phys., 12, 8553-8566, 10.5194/acp-12-8553-2012, 2012.

Graber, E. R. and Rudich, Y.: Atmospheric HULIS: How humic-like are they? A comprehensive and critical review, Atmos. Chem. Phys., 6, 729-753, 10.5194/acp-6-729-2006, 2006.

Grythe, H., Ström, J., Krejci, R., Quinn, P., and Stohl, A.: A review of sea-spray aerosol source functions using a large global set of sea salt aerosol concentration measurements, Atmospheric Chemistry and Physics, 14, 1277-1297, 10.5194/acp-14-1277-2014, 2014.

Gu, W., Xie, Z., Wei, Z., Chen, A., Jiang, B., Yue, F., and Yu, X.: Marine Fresh Carbon Pool Dominates Summer Carbonaceous Aerosols Over Arctic Ocean, J. Geophys. Res., [Atmos.], 128, 10.1029/2022jd037692, 2023.

Gupta, M., Sahu, L. K., Tripathi, N., Sudheer, A. K., and Singh, A.: Processes Controlling DMS Variability in Marine Boundary Layer of the Arabian Sea During Post-Monsoon Season of 2021, J. Geophys. Res., [Atmos.], 130, 10.1029/2024jd042547, 2025.

Hedges, J. I.: Global biogeochemical cycles: progress and problems, Marine Chemistry, 39, 67-93, https://doi.org/10.1016/0304-4203(92)90096-S, 1992.

Hersbach, H., Bell, B., Berrisford, P., Hirahara, S., Horányi, A., Muñoz-Sabater, J., Nicolas, J., Peubey, C., Radu, R., Schepers, D., Simmons, A., Soci, C., Abdalla, S., Abellan, X., Balsamo, G., Bechtold, P., Biavati, G., Bidlot, J., Bonavita, M., De Chiara, G., Dahlgren, P., Dee, D., Diamantakis, M., Dragani, R., Flemming, J., Forbes, R., Fuentes, M., Geer, A., Haimberger, L., Healy, S., Hogan, R. J., Hólm, E., Janisková, M., Keeley, S., Laloyaux, P., Lopez, P., Lupu, C., Radnoti, G., de Rosnay, P., Rozum, I., Vamborg, F., Villaume, S., and Thépaut, J. N.: The ERA5 global reanalysis, Quarterly Journal of the Royal Meteorological Society, 146, 1999-2049, 10.1002/qj.3803, 2020.

Hoque, M., Kawamura, K., Seki, O., and Hoshi, N.: Spatial distributions of dicarboxylic acids, ω-oxoacids, pyruvic acid and α-dicarbonyls in the remote marine aerosols over the North Pacific, Marine Chemistry, 172, 1-11, 10.1016/j.marchem.2015.03.003, 2015.

Hoque, M. M. M., Kawamura, K., and Uematsu, M.: Spatio-temporal distributions of dicarboxylic acids, ω-oxocarboxylic acids, pyruvic acid, α-dicarbonyls and fatty acids in the marine aerosols from the North and South Pacific, Atmospheric Research, 185, 158-168, 10.1016/j.atmosres.2016.10.022, 2017.

Hsu, S.-C., Wong, G. T. F., Gong, G.-C., Shiah, F.-K., Huang, Y.-T., Kao, S.-J., Tsai, F., Candice Lung, S.-C., Lin, F.-J., Lin, I. I., Hung, C.-C., and Tseng, C.-M.: Sources, solubility, and dry deposition of aerosol trace elements over the East China Sea, Mar. Chem., 120, 116-127, 10.1016/j.marchem.2008.10.003, 2010.

Hu, J., Li, J., Tsona Tchinda, N., Song, Y., Xu, M., Li, K., and Du, L.: Underestimated role of sea surface temperature in sea spray aerosol formation and climate effects, npj Climate and Atmospheric Science, 7, 10.1038/s41612-024-00823-x, 2024.

Huang, S., Wu, Z., Poulain, L., van Pinxteren, M., Merkel, M., Assmann, D., Herrmann, H., and Wiedensohler, A.: Source apportionment of the organic aerosol over the Atlantic Ocean from 53° N to 53° S: significant contributions from marine emissions and long-range transport, Atmos. Chem. Phys., 18, 18043-18062, 10.5194/acp-18-18043-2018, 2018.

Huang, S., Wu, Z., Wang, Y., Poulain, L., Hopner, F., Merkel, M., Herrmann, H., and Wiedensohler, A.: Aerosol Hygroscopicity and its Link to Chemical Composition in a Remote Marine Environment Based on Three Transatlantic Measurements, Environmental Science & Technology, 56, 9613-9622, 10.1021/acs.est.2c00785, 2022.

Huebert, B. J. and Charlson, R. J.: Uncertainties in data on organic aerosols, Tellus B: Chemical and Physical Meteorology, 52, 10.3402/tellusb.v52i5.17099, 2000.

Kunwar, B. and Kawamura, K.: One-year observations of carbonaceous and nitrogenous components and major ions in the aerosols from subtropical Okinawa Island, an outflow region of Asian dusts, Atmos. Chem. Phys., 14, 1819-1836, 10.5194/acp-14-1819-2014, 2014.

Laskin, A., Laskin, J., and Nizkorodov, S. A.: Chemistry of atmospheric brown carbon, Chem. Rev., 115, 4335-4382, 10.1021/cr5006167, 2015.

Lawler, M. J., Lewis, S. L., Russell, L. M., Quinn, P. K., Bates, T. S., Coffman, D. J., Upchurch, L. M., and Saltzman, E. S.: North Atlantic marine organic aerosol characterized by novel offline thermal desorption mass spectrometry: polysaccharides, recalcitrant material, and secondary organics, Atmospheric Chemistry and Physics, 20, 16007-16022, 10.5194/acp-20-16007-2020, 2020.

Lewis, E. and Schwartz, S.: Sea Salt Aerosol Production: Mechanisms, Methods, Measurements and Models—A Critical Review, Washington DC American Geophysical Union Geophysical Monograph Series, 152, 3719, 10.1029/GM152, 2004.

Li, J., Han, Z., Fu, P., Yao, X., and Liang, M.: Seasonal characteristics of emission, distribution, and radiative effect of marine organic aerosols over the western Pacific Ocean: an investigation with a coupled regional climate aerosol model, Atmospheric Chemistry and Physics, 24, 3129-3161, 10.5194/acp-24-3129-2024, 2024.

Lim, H. J. and Turpin, B. J.: Origins of primary and secondary organic aerosol in Atlanta: results of time-resolved measurements during the Atlanta Supersite Experiment, Environ. Sci. Technol., 36, 4489-4496, 10.1021/es0206487, 2002.

Ma, X., Li, K., Zhang, S., Tchinda, N. T., Li, J., Herrmann, H., and Du, L.: Molecular characteristics of sea spray aerosols during aging with the participation of marine volatile organic compounds, Science of the Total Environment, 954, 176380, 10.1016/j.scitotenv.2024.176380, 2024.

Miyazaki, Y., Kawamura, K., and Sawano, M.: Size distributions and chemical characterization of water-soluble organic aerosols over the western North Pacific in summer, Journal of Geophysical Research: Atmospheres, 115, 10.1029/2010jd014439, 2010.

Miyazaki, Y., Suzuki, K., Tachibana, E., Yamashita, Y., Muller, A., Kawana, K., and Nishioka, J.: New index of organic mass enrichment in sea spray aerosols linked with senescent status in marine phytoplankton, Scientific Reports, 10, 17042, 10.1038/s41598-020-73718-5, 2020.

Miyazaki, Y., Yamashita, Y., Kawana, K., Tachibana, E., Kagami, S., Mochida, M., Suzuki, K., and Nishioka, J.: Chemical transfer of dissolved organic matter from surface seawater to sea spray water-soluble organic aerosol in the marine atmosphere, Scientific Reports, 8, 14861, 10.1038/s41598-018-32864-7, 2018a.

Miyazaki, Y., Yamashita, Y., Kawana, K., Tachibana, E., Kagami, S., Mochida, M., Suzuki, K., and Nishioka, J.: Chemical transfer of dissolved organic matter from surface seawater to sea spray water-soluble organic aerosol in the marine atmosphere, Sci. Rep., 8, 14861, 10.1038/s41598-018-32864-7, 2018b.

Murphy, K. R., Stedmon, C. A., Graeber, D., and Bro, R.: Fluorescence spectroscopy and multi-way techniques. PARAFAC, Analytical Methods, 5, 10.1039/c3ay41160e, 2013.

O'Dowd, C. D., Langmann, B., Varghese, S., Scannell, C., Ceburnis, D., and Facchini, M. C.: A combined organic-inorganic sea-spray source function, Geophysical Research Letters, 35, 10.1029/2007gl030331, 2008.

O'Dowd, C. D., Facchini, M. C., Cavalli, F., Ceburnis, D., Mircea, M., Decesari, S., Fuzzi, S., Yoon, Y. J., and Putaud, J. P.: Biogenically driven organic contribution to marine aerosol, Nature, 431, 676-680, 10.1038/nature02959, 2004.

Pöhlker, C., Huffman, J. A., and Pöschl, U.: Autofluorescence of atmospheric bioaerosols – fluorescent biomolecules and potential interferences, Atmos. Meas. Tech., 5, 37-71, 10.5194/amt-5-37-2012, 2012.

Prather, K. A., Bertram, T. H., Grassian, V. H., Deane, G. B., Stokes, M. D., Demott, P. J., Aluwihare, L. I., Palenik, B. P., Azam, F., Seinfeld, J. H., Moffet, R. C., Molina, M. J., Cappa, C. D., Geiger, F. M., Roberts, G. C., Russell, L. M., Ault, A. P., Baltrusaitis, J., Collins, D. B., Corrigan, C. E., Cuadra-Rodriguez, L. A., Ebben, C. J., Forestieri, S. D., Guasco, T. L., Hersey, S. P., Kim, M. J., Lambert, W. F., Modini, R. L., Mui, W., Pedler, B. E., Ruppel, M. J., Ryder, O. S., Schoepp, N. G., Sullivan, R. C., and Zhao, D.: Bringing the ocean into the laboratory to probe the chemical complexity of sea spray aerosol, Proc. Natl. Acad. Sci. USA, 110, 7550-7555, 10.1073/pnas.1300262110, 2013.

Quinn, P. K. and Bates, T. S.: The case against climate regulation via oceanic phytoplankton sulphur emissions, Nature, 480, 51-56, 10.1038/nature10580, 2011.

Quinn, P. K., Coffman, D. J., Johnson, J. E., Upchurch, L. M., and Bates, T. S.: Small fraction of marine cloud condensation nuclei made up of sea spray aerosol, Nat. Geosci., 10, 674-679, 10.1038/ngeo3003, 2017.

Quinn, P. K., Collins, D. B., Grassian, V. H., Prather, K. A., and Bates, T. S.: Chemistry and related properties of freshly emitted sea spray aerosol, Chem. Rev., 115, 4383-4399, 10.1021/cr500713g, 2015a.

Quinn, P. K., Collins, D. B., Grassian, V. H., Prather, K. A., and Bates, T. S.: Chemistry and related properties of freshly emitted sea spray aerosol, Chemical Review, 115, 4383-4399, 10.1021/cr500713g, 2015b.

Quinn, P. K., Bates, T. S., Schulz, K. S., Coffman, D. J., Frossard, A. A., Russell, L. M., Keene, W. C., and Kieber, D. J.:
Contribution of sea surface carbon pool to organic matter enrichment in sea spray aerosol, Nat. Geosci., 7, 228-232,
10.1038/ngeo2092, 2014.
Rinaldi, M., Fuzzi, S., Decesari, S., Marullo, S., Santoleri, R., Provenzale, A., von Hardenberg, J., Ceburnis, D., Vaishya, A.,
O'Dowd, C. D., and Facchini, M. C.: Is chlorophyll-athe best surrogate for organic matter enrichment in submicron primary
marine aerosol?, Journal of Geophysical Research: Atmospheres, 118, 4964-4973, 10.1002/jgrd.50417, 2013.
Russell, L. M., Hawkins, L. N., Frossard, A. A., Quinn, P. K., and Bates, T. S.: Carbohydrate-like composition of submicron
atmospheric particles and their production from ocean bubble bursting, The Proceedings of the National Academy of
Sciences 107, 6652-6657, 10.1073/pnas.0908905107, 2010.
Sahu, L. K., Kondo, Y., Miyazaki, Y., Kuwata, M., Koike, M., Takegawa, N., Tanimoto, H., Matsueda, H., Yoon, S. C., and
Kim, Y. J.: Anthropogenic aerosols observed in Asian continental outflow at Jeju Island, Korea, in spring 2005, J. Geophys.
Res., [Atmos.], 114, 10.1029/2008jd010306, 2009.
Salter, M. E., Nilsson, E. D., Butcher, A., and Bilde, M.: On the seawater temperature dependence of the sea spray aerosol
generated by a continuous plunging jet, Journal of Geophysical Research: Atmospheres, 119, 9052-9072,
10.1002/2013jd021376, 2014.
Santander, M. V., Schiffer, J. M., Lee, C., Axson, J. L., Tauber, M. J., and Prather, K. A.: Factors controlling the transfer of
biogenic organic species from seawater to sea spray aerosol, Scientific Reports, 12, 3580, 10.1038/s41598-022-07335-9,
685    2022.
Santander, M. V., Mitts, B. A., Pendergraft, M. A., Dinasquet, J., Lee, C., Moore, A. N., Cancelada, L. B., Kimble, K. A.,
Malfatti, F., and Prather, K. A.: Tandem Fluorescence Measurements of Organic Matter and Bacteria Released in Sea Spray
Aerosols, Environmental Science & Technology, 55, 5171-5179, 10.1021/acs.est.0c05493, 2021.
Schmitt-Kopplin, P., Liger-Belair, G., Koch, B. P., Flerus, R., Kattner, G., Harir, M., Kanawati, B., Lucio, M., Tziotis, D.,
Hertkorn, N., and Gebefügi, I.: Dissolved organic matter in sea spray: a transfer study from marine surface water to aerosols,
Biogeosciences, 9, 1571-1582, 10.5194/bg-9-1571-2012, 2012.
Shank, L. M., Howell, S., Clarke, A. D., Freitag, S., Brekhovskikh, V., Kapustin, V., McNaughton, C., Campos, T., and
Wood, R.: Organic matter and non-refractory aerosol over the remote Southeast Pacific: oceanic and combustion sources,
Atmos. Chem. Phys., 12, 557-576, 10.5194/acp-12-557-2012, 2012.
Siemer, J. P., Machín, F., González-Vega, A., Arrieta, J. M., Gutiérrez-Guerra, M. A., Pérez-Hernández, M. D., Vélez-Belchí,
P., Hernández-Guerra, A., and Fraile-Nuez, E.: Recent Trends in SST, Chl-a, Productivity and Wind Stress in Upwelling and
Open Ocean Areas in the Upper Eastern North Atlantic Subtropical Gyre, J. Geophys. Res., [Oceans], 126,
10.1029/2021jc017268, 2021.
Sinclair, K., van Diedenhoven, B., Cairns, B., Alexandrov, M., Moore, R., Ziemba, L. D., and Crosbie, E.: Observations of
Aerosol-Cloud Interactions During the North Atlantic Aerosol and Marine Ecosystem Study, Geophysical Research Letters,
47, 10.1029/2019gl085851, 2020.
Spracklen, D. V., Arnold, S. R., Sciare, J., Carslaw, K. S., and Pio, C.: Globally significant oceanic source of organic carbon
aerosol, Geophys. Res. Lett., 35, 10.1029/2008gl033359, 2008.
Stedmon, C. A. and Bro, R.: Characterizing dissolved organic matter fluorescence with parallel factor analysis: a tutorial,
Limnology and Oceanography: Methods, 6, 572-579, 10.4319/lom.2008.6.572, 2008.
Tang, J., Xu, B., Zhao, S., Li, J., Tian, L., Geng, X., Jiang, H., Mo, Y., Zhong, G., Jiang, B., Chen, Y., Tang, J., and Zhang,
G.: Long-Emission-Wavelength Humic-Like Component (L-HULIS) as a Secondary Source Tracer of Brown Carbon in the
Atmosphere, Journal of Geophysical Research: Atmospheres, 129, 10.1029/2023jd040144, 2024.
Tripathi, N., Girach, I. A., Kompalli, S. K., Murari, V., Nair, P. R., Babu, S. S., and Sahu, L. K.: Sources and Distribution of
Light NMHCs in the Marine Boundary Layer of the Northern Indian Ocean During Winter: Implications to Aerosol
Formation, J. Geophys. Res., [Atmos.], 129, 10.1029/2023jd039433, 2024.
Tripathi, N., Sahu, L. K., Singh, A., Yadav, R., Patel, A., Patel, K., and Meenu, P.: Elevated Levels of Biogenic Nonmethane
Hydrocarbons in the Marine Boundary Layer of the Arabian Sea During the Intermonsoon, J. Geophys. Res., [Atmos.], 125,
10.1029/2020jd032869, 2020.
Trueblood, J. V., Wang, X., Or, V. W., Alves, M. R., Santander, M. V., Prather, K. A., and Grassian, V. H.: The Old and the
New: Aging of Sea Spray Aerosol and Formation of Secondary Marine Aerosol through OH Oxidation Reactions, ACS
Earth and Space Chemistry, 3, 2307-2314, 10.1021/acsearthspacechem.9b00087, 2019.

Tuchen, F. P., Perez, R. C., Foltz, G. R., Brandt, P., Subramaniam, A., Lee, S. K., Lumpkin, R., and Hummels, R.: Modulation of Equatorial Currents and Tropical Instability Waves During the 2021 Atlantic Niño, J. Geophys. Res., [Oceans], 129, 10.1029/2023jc020431, 2023.

Turpin, B. J. and Huntzicker, J. J.: Identification of Secondary Organic Aerosol Episodes and Quantitation of Primary and Secondary Organic Aerosol Concentrations During SCAQS, Atmospheric Environment, 29, 3527-3544, 10.1016/1352-2310(94)00276-Q, 1995.

Vergara-Temprado, J., Murray, B. J., Wilson, T. W., O'Sullivan, D., Browse, J., Pringle, K. J., Ardon-Dryer, K., Bertram, A. K., Burrows, S. M., Ceburnis, D., DeMott, P. J., Mason, R. H., O'Dowd, C. D., Rinaldi, M., and Carslaw, K. S.: Contribution of feldspar and marine organic aerosols to global ice nucleating particle concentrations, Atmospheric Chemistry and Physics, 17, 3637-3658, 10.5194/acp-17-3637-2017, 2017.

Vignati, E., Facchini, M. C., Rinaldi, M., Scannell, C., Ceburnis, D., Sciare, J., Kanakidou, M., Myriokefalitakis, S., Dentener, F., and O'Dowd, C. D.: Global scale emission and distribution of sea-spray aerosol: Sea-salt and organic enrichment, Atmos. Environ., 44, 670-677, 10.1016/j.atmosenv.2009.11.013, 2010.

Wang, J., Zhang, H. H., Booge, D., Zhang, Y. Q., Li, X. J., Wu, Y. C., Zhang, J. W., and Chen, Z. H.: Isoprene Production and Its Driving Factors in the Northwest Pacific Ocean, Global Biogeochem. Cy., 37, 10.1029/2023gb007841, 2023a.

Wang, X., Deane, G. B., Moore, K. A., Ryder, O. S., Stokes, M. D., Beall, C. M., Collins, D. B., Santander, M. V., Burrows, S. M., Sultana, C. M., and Prather, K. A.: The role of jet and film drops in controlling the mixing state of submicron sea spray aerosol particles, Proc. Natl. Acad. Sci. USA, 114, 6978-6983, 10.1073/pnas.1702420114, 2017.

Wang, X., Sultana, C. M., Trueblood, J., Hill, T. C., Malfatti, F., Lee, C., Laskina, O., Moore, K. A., Beall, C. M., McCluskey, C. S., Cornwell, G. C., Zhou, Y., Cox, J. L., Pendergraft, M. A., Santander, M. V., Bertram, T. H., Cappa, C. D., Azam, F., DeMott, P. J., Grassian, V. H., and Prather, K. A.: Microbial Control of Sea Spray Aerosol Composition: A Tale of Two Blooms, ACS Central Science, 1, 124-131, 10.1021/acscentsci.5b00148, 2015.

Wang, Y., Zhang, P., Li, J., Liu, Y., Zhang, Y., Li, J., and Han, Z.: An updated aerosol simulation in the Community Earth System Model (v2.1.3): dust and marine aerosol emissions and secondary organic aerosol formation, Geoscientific Model Development, 17, 7995-8021, 10.5194/gmd-17-7995-2024, 2024.

Wang, Y., Zhang, Y., Li, W., Wu, G., Qi, Y., Li, S., Zhu, W., Yu, J. Z., Yu, X., Zhang, H. H., Sun, J., Wang, W., Sheng, L., Yao, X., Gao, H., Huang, C., Ma, Y., and Zhou, Y.: Important roles and formation of atmospheric organosulfates in marine organic aerosols: Influence of phytoplankton emissions and anthropogenic pollutants, Environ. Sci. Technol., 57, 10284-10294, 10.1021/acs.est.3c01422, 2023b.

Wolf, M. J., Coe, A., Dove, L. A., Zawadowicz, M. A., Dooley, K., Biller, S. J., Zhang, Y., Chisholm, S. W., and Cziczo, D. J.: Investigating the Heterogeneous Ice Nucleation of Sea Spray Aerosols Using Prochlorococcus as a Model Source of Marine Organic Matter, Environmental Science & Technology, 53, 1139-1149, 10.1021/acs.est.8b05150, 2019.

Xu, W., Ovadnevaite, J., Fossum, K. N., Lin, C., Huang, R.-J., Ceburnis, D., and O'Dowd, C.: Sea spray as an obscured source for marine cloud nuclei, Nature Geoscience, 15, 282-286, 10.1038/s41561-022-00917-2, 2022.

Yu, Y., Wang, H., Wang, T., Song, K., Tan, T., Wan, Z., Gao, Y., Dong, H., Chen, S., Zeng, L., Hu, M., Wang, H., Lou, S., Zhu, W., and Guo, S.: Elucidating the importance of semi-volatile organic compounds to secondary organic aerosol formation at a regional site during the EXPLORE-YRD campaign, Atmos. Environ., 246, 10.1016/j.atmosenv.2020.118043, 2021.

Yu, Z. and Li, Y.: Marine volatile organic compounds and their impacts on marine aerosol-A review, Sci. Total Environ., 768, 145054, 10.1016/j.scitotenv.2021.145054, 2021.

Zhang, W. and Gu, D.: Geostationary satellite reveals increasing marine isoprene emissions in the center of the equatorial Pacific Ocean, npj Climate and Atmospheric Science, 5, 1, 10.1038/s41612-022-00311-0, 2022.

Zhang, Y., Wang, Y., Li, S., Yi, Y., Guo, Y., Yu, C., Jiang, Y., Ni, Y., Hu, W., Zhu, J., Qi, J., Shi, J., Yao, X., and Gao, H.: Sources and Optical Properties of Marine Organic Aerosols Under the Influence of Marine Emissions, Asian Dust, and Anthropogenic Pollutants, J. Geophys. Res., [Atmos.], 130, 10.1029/2025jd043472, 2025.

Zhao, X., Liu, X., Burrows, S. M., and Shi, Y.: Effects of marine organic aerosols as sources of immersion-mode ice-nucleating particles on high-latitude mixed-phase clouds, Atmospheric Chemistry and Physics, 21, 2305-2327, 10.5194/acp-21-2305-2021, 2021.