# Peer review of "Biogenically driven marine organic aerosol production over the West"

_EGUsphere, 2025_

## Author Response (AR2)

Dear Prof. Manmohan Sarin and reviewers,

We appreciate all your detailed and valuable suggestions on our manuscript (egusphere-2025-3951). We have carefully considered the comments and revised the manuscript accordingly. Please see the point-by-point response below and changes are marked blue in the revised manuscript.

Thanks very much!

Most sincerely,

Yujue Wang and Wei Xu on behalf of all co-authors

**Point-by-point response to review comments**

**Note**: *Review comments are in italicized font*. Our responses are indented and in normal font. Revised text in the manuscript or supplementary is in blue color.

**Referee #1**

*This study presents a very detailed analysis of the organic aerosol fraction in the marine atmosphere. The author made efforts to try to find a better way of parameterizing the primary and secondary OC in marine aerosols using common measurements of seawater (chl-a and Na+). However, it is not clear about the main conclusion of applying the proposed new method and its performance. Key gaps in interpreting the fluorescent components are missing. I expect further elaborations on these issues together with other major comments. Please see the following comments for details.*

> **Response**: Thank you very much for your overall positive comments on our manuscript. We appreciate your detailed comments regarding the performance of the proposed parameterization and the interpretation of the fluorescent components, which we have carefully addressed and revised the manuscript according to the following comments. For the different contributions of MPOC identified by the mass concentrations and the fluorescent properties, we have elaborated in the *major comment 10* and revised the manuscript accordingly.

**Major comments:**

*The authors define marine primary organic carbon MPOC as those emitted from marine bubble bursting. How about those emitted by other processes like wave breaking? The definition of MSOC is even unclear. In the Abstract, it is defined as "secondary organic carbon (MSOC) formed via gas-to-particle conversion remains poorly quantified". Are these gases those only emitted from the ocean?*
*How about the primary OC and gas precursors transported from the terrestrial environments?*
*In summary, it is unclear what does "marine" mean here. Does it only refer to marine atmosphere?*

> **Response**: Thanks for pointing out the unclear definition. In this study, marine primary organic carbon (MPOC) is the organic carbon in the sea spray aerosols, which includes those emitted from bubble bursting and wave breaking. Marine secondary organic carbon (MSOC) includes the organic aerosols formed via gas-to-particle conversion of gaseous precursors and via the oxidation/aging processes of primary organic aerosols. The gaseous precursors and aged organic aerosols could be from both ocean emissions and long-range transported from terrestrial environments. We have defined MPOC and MSOC in the abstract and the main text (lines 19−21, 236−238, 252−253).

We added the back trajectory analysis of air masses (Fig. S1) to analyze the influence of terrestrial outflows. The air masses during the cruises were mainly transported from open oceanic regions, and thus the impacts of terrestrial outflows were limited. Related descriptions on the potential impacts of transported terrestrial air masses are added in the revised version (lines 164−166, 315−324).

Yes, "marine" here refers to marine atmospheres. Atmospheric aerosols over the oceans could be from diverse sources, including primary and secondary marine emissions, and transported pollutants (Brooks and Thornton, 2018). We cannot exclude the potential impacts of transported terrestrial compounds based on the cruise observations. The secondarily-formed marine organic aerosols in this work could be sourced both from marine emissions and transported from terrestrial environments.

**Lines 19−21 in the abstract:**

However, the abundance of marine primary organic carbon (MPOC) generated by sea spray and secondary organic carbon (MSOC) formed via gas-to-particle conversion or atmospheric oxidation/aging processes remains poorly quantified, which hinders our understanding on the climate effects of marine aerosols.

**Lines 236−238:**

Based on the correlation analysis of the observed parameters, we proposed a parameterization scheme to separate the marine primarily-emitted OC (MPOC) in the SSA generated through wave breaking or bubble bursting processes and the secondarily formed organic carbon (MSOC) in the marine aerosols over WPO.

**Lines 252−253:**

The MSOC here includes the organic aerosols formed via gas-to-particle conversion of gaseous precursors and oxidation/aging processes of primary OC.

**Lines 164−166:**

The air masses were mainly transported from open oceanic regions, and thus the impacts of terrestrial outflows were limited during the cruises (Fig. S1).

**Lines 315−324:**

It is noted that, based on the shipboard in-situ observation, we cannot exclude the potential impacts of gaseous precursors or aged organic aerosols long-range transported from terrestrial environments, which were mostly in the MSOC fraction. The organic aerosols transported from terrestrial environments were secondary or aged organic aerosols, and tend to be water-soluble organic compounds (Boreddy et al., 2018; De Jonge et al., 2024; Miyazaki et al., 2010). Based on the air mass back trajectories (Fig.

S1) and the weak correlations between OC and EC stated in section 3.1, the impacts of transported continental outflows were limited during the cruises.

**Newly added Figure S1:**

[Figure]

[Figure]

Figure S1 The 24-hr back trajectories of air masses during the cruises in (a) spring and (b) summer.

*1. L115 "Organic matters were the dominant components in the fine particles, which respectively contributed 18%−75% (40% on average) and 13%−74% (48% on average) of the PM2.5 mass in spring and summer." The authors measured the mass concentrations of organic matters and inorganic ions. In addition to these measured composition in PM2.5, how about the contribution of other composition, e.g., other metal elements, ions, elemental carbon, in the mass of PM2.5?*

**Response**: In this study, we measured organic carbon (organic matter), water-soluble ions ($Na^+$, $NH_4^+$, $K^+$, $Mg^{2+}$, $Ca^{2+}$, $Cl^-$, $NO_3^-$ and $SO_4^{2-}$), and elemental carbon (EC), which were summed as the total mass of $PM_{2.5}$. The average EC concentrations were 0.066 and 0.055 µgC m$^{-3}$ during the spring and the summer campaigns, of which the contributions were much lower than OM. Previous studies have suggested that organics and water-soluble ions are the major contributors to the marine aerosol mass. Other metal elements (e.g., Al, Fe, Ti, Sr, Ba, Mn, etc.) accounted for <3.5% of the marine aerosol mass based on the measurements over the East China Sea (Hsu et al., 2010). Related descriptions have been added in lines 130−132, 141−142.

We agree with the reviewer that metal elements (e.g., Fe, Al, etc.) are also important in marine aerosols. We did not analyze metal elements in this work due to the limited mass loading of the collected marine aerosol samples. We will try to measure the metal elements in our future studies.

**Lines 130−132:**

The EC concentrations were 0.066 ± 0.056 µgC m$^{-3}$ and 0.055 ± 0.052 µgC m$^{-3}$ during the spring and the summer observations, much lower than those observed over coastal areas typically influenced by continental outflows (Sahu et al., 2009; Zhang et al., 2025).

**Lines 140−143:**

The mass concentrations of PM$_{2.5}$ were calculated by summing the measured OM, EC, and water-soluble ions. Metal elements were not measured in this study, which contributed <3.5% of the marine aerosol mass concentration over the East China Sea (Hsu et al., 2010). Without considering the metal elements, we may overestimate the organic proportion in marine aerosols.

2. *L118-122: how are the sampling and measurement uncertainties of OC in PM2.5 and TSP?*
*What could be the reasons that "organic fractions were dominant in the submicron marine aerosols"? Could these reasons be consistent with the main sources of MPOC of MSOC?*

**Response**: The sampling and measurement uncertainties of OC in the aerosol samples are described in lines 143−147.

Sea spray aerosols (SSA) or primary marine aerosols could be formed by film drops and jet drops during wave breaking and bubble-bursting processes. Submicron SSA are mainly formed by film drops produced from bursting bubble-cap film, which is enriched with hydrophobic organic matters contained within the sea surface microlayer (Burrows et al., 2014; Wang et al., 2017). In contrast, jet drops formed from the base of bursting bubbles are mainly produce larger supermicron particles from bulk seawater, which comprises sea salts and smaller fraction of organics (Wang et al., 2017). Thus, the dominance of organics in submicron marine aerosols was due to the enriched organic compounds in the air−water interface, which could be efficiently transferred into the submicron sea spray aerosols through film drops during wave breaking and bubble-bursting processes. This is now added in lines 138−140, 232−234.

The reason is consistent with the main source of MPOC related to marine biological activity, indicated by seawater *Chl-a*. Marine phytoplankton could produce gel-like aggregates and contribute to extracellular polymer particles, water-insoluble polysaccharide-containing transparent exopolymer, and protein-containing organics, etc. in seawater (Aller et al., 2017; Lawler et al., 2020). These hydrophobic organics could be enriched in the surface seawater and easily transferred into the submicron aerosols over the ocean by film drops. Related descriptions have been added in lines 182−185.

**Lines 138−140:**

Film drops could efficiently transfer hydrophobic organic compounds enriched in the air−water interface into the submicron aerosols, which explained the size-selective enrichment of organics in marine aerosols (Cochran et al., 2016; Prather et al., 2013; Quinn et al., 2015; Wang et al., 2017).

**Lines 143−147:**

During the sampling, positive artifacts of OC may exist due to the absorption of gaseous organic vapor on the filters, and negative artifacts may exist due to the evaporation of volatile organic compounds (Huebert and Charlson, 2000). The OC concentration was measured using thermal-optical analysis. Quantification uncertainty may be introduced due to the formation of pyrolyzed OC, which complicates the accurate determination of the OC/EC split point (Cao et al., 2025; Chow et al., 2004).

**Lines 182−185:**

Marine phytoplankton could produce gel-like aggregates and contribute to extracellular polymer particles, water-insoluble polysaccharide-containing transparent exopolymer, and protein-containing organics, etc. in seawater (Aller et al., 2017; Lawler et al., 2020). These organic substances could be enriched in the surface seawater and then transferred into the atmospheric aerosols within the marine boundary layer.

**Lines 232−234:**

Organic matters in marine aerosols are enriched in the submicron SSA, which is mainly formed by film drops from bursting bubble-cap films (Wang et al., 2017). In contrast, the majority of the sea salts exist in larger supermicron or coarse-mode particles generated by jet drops from the base of bursting bubbles (Wang et al., 2017).

3. *L177-179: is the representativeness of Na+ of overall SSA production also useful in coastal marine atmosphere considering the anthropogenic/terrestrial sources of Na+? The conclusion is derived based on PM$_{2.5}$ samples. Many marine studies collect TSP or PM$_{10}$ samples. For these samples, can we interpret the Na+ measurements in this way as well?*

**Response**: Thanks for reminding the potential sources of Na$^+$ from terrestrial sources. We have conducted cruise observations of the aerosols in coastal marine atmospheres over the East Asian marginal seas. Among the marine aerosol sources resolved by PMF source apportionment, Na$^+$ mainly existed in sea spray aerosols and dust aerosols (Figure R1), and the Na$^+$ from anthropogenic sources are neglectable (Zhang et al., 2025). When using Na$^+$ as the indicator of SSA production, Na$^+$ transported by terrestrial dust storms should be excluded. Related description has been added in lines 227−230.

For the collected TSP samples, the OC concentrations did not display obvious correlation with the seawater *Chl-a* due to different production processes of OC and sea salts. We added related discussion in lines 230−234 and Fig. S4. The Na$^+$ in the PM$_{10}$ or TSP samples could be used as an indicator of the bulk SSA production. However, it might not be a good input to estimate the organics from SSA.

[Figure]

Figure R1 Explained variation of the marine aerosol source factors apportioned by PMF model over East Asian marginal seas (Figure S7 in Zhang et al., 2025)

**Lines 227−230:**

It should be noted that dust storms also transport $Na^+$ to marine atmospheres, especially over the marginal seas (Zhang et al., 2025). When using $Na^+$ in marine aerosols as the indicator of SSA production, the $Na^+$ contributed by transported dust storms should be excluded, especially during dust episodes.

**Lines 230−234:**

For the collected TSP samples, the OC concentrations did not display an obvious correlation with the seawater *Chl-a* (Fig. S4). This is because the dominant production processes of OC and sea salts are different. Organic matters in marine aerosols are enriched in the submicron SSA, which is mainly formed by film drops from bursting bubble-cap films (Wang et al., 2017). In contrast, the majority of the sea salt mass exist in larger supermicron or coarse-mode particles generated by jet drops from the base of bursting bubbles (Wang et al., 2017).

**Newly added Figure S4:**

[Figure]

Figure S4 Scatter plot of the OC concentration in the collected TSP samples and seawater *Chl-a* during the cruises.

4. *L244-246 "The estimated MSOC matched better with the WSOC in the marine aerosols when using a combination of [Chl-a] and [Na+] as the input parameters and considering the variation of sea spray aerosols (Fig. 3c, 3g)". I don't see the evidence why the estimated MSOC matched better with the WSOC when using a combination of [Chl-a] and [Na+] and considering the variation of sea spray aerosols. The "r" value in Fig. 3c and 3g is the same (0.73). Please further explain.*

**Response**: The performance of the parameterization was evaluated based on the fitting line slope and correlation coefficient (r) of WSOC and estimated MSOC. It means that the estimated MPOC shows a similar variation trend to WIOC if with a r value closer to 1, and a good comparison with the WIOC mass concentrations if with a fitting line slope closer to 1. The fitting line slope of WSOC and estimated MSOC was closer to 1 when using equation 1 and 3, and more estimated MSOC concentrations fall within the WSOC/MSOC 3:1 and 1:3 lines compared with the results based on equations 1 and 2. We now have added related descriptions to be clear (lines 302−305, 315−318).

**Lines 302−305:**

Both the correlation coefficients (r) and the slopes of the fitting line between WIOC and estimated MPOC are used to evaluate the performance of different MPOC parameterization approaches. It means that the estimated MPOC shows a similar variation trend to WIOC if with a r value closer to 1, and a good comparison with the WIOC mass concentrations if with a fitting line slope closer to 1.

**Lines 315−318:**

Based on equations 1 and 3, the estimated MSOC concentrations in half of the samples fall within the WSOC/MSOC 3:1 and 1:3 lines, and the fitting line slope (0.50) was closer to 1 (Fig. 3h). Using equations 1 and 2, the fitting line slope of WSOC and estimated MSOC was 0.46, and 46% of the estimated MSOC concentrations fall within the WSOC/MSOC 3:1 and 1:3 lines (Fig. 3c).

5. *L248 and Fig. 3d & 3h: how about WSOC? Can you compare the estimated MSOC with those estimated by the parameters/methods of Gantt and Vignati?*

**Response**: The comparisons of WSOC and the estimated MSOC using the methods from Gantt et al. (2011) and Vignati et al. (2010) are now added in Figure 3 and the main text (lines 332−335).

**Lines 332−335:**

The estimated MSOC using the parameterizations from Gantt et al. (2011) or Vignati et al. (2010) showed similar variation trends to the WSOC in the collected aerosols samples. The comparison of the estimated MSOC and the WSOC concentrations using formulations in literatures (slopes in Fig. 3e, 3j), however, were not as good as those estimated in this study (slopes in Fig. 3h).

**Revised Figure 3 (panels e and j are added):**

[Figure]

Figure 3 The scatter plots of OC in marine aerosols as a function of (a) seawater [*Chl-a*] and (f) ([*Chl-a*] × [Na$^+$]$^{0.45}$) during the two cruises; (b, g) Comparison of WIOC and the estimated MPOC based on the regression in panel (a) and panel (f); (c, h) Comparison of WSOC and the estimated MSOC; (d, i) Comparison of WIOC and the estimated MPOC, and (e, j) Comparison of WSOC and the estimated MSOC using the formulation of Vignati (2010) and Gantt (2011). The dashed lines in panels (a, f) are the regression line of [OC] and [*Chl-a*] or ([*Chl-a*] × [Na$^+$]$^{0.45}$) with 0–30% percentile ratios, indicated by solid markers, during Cruise I (blue) and Cruise II (red). The regressions line in panels (b−e, g−j) represent the correlation between WIOC and the estimated MPOC or between WSOC and the estimated MSOC in each panel during the two cruises.

6. *L253-254 "These parameterizations perform well to trace the variation trends of MPOC. However, they might lead to an underestimation of the primary MOA over the Northwest Pacific Ocean." What could be reasons? Is the difference between MPOC and WIOC can (partly) explain the different performance? Can location/terrestrial aerosols play a role in the difference? Please further explain/discuss.*

**Response**: Thanks for the suggestions. We agree with the review that locations could be an important reason for the difference, and have added the description in underline 338−340. The seawater compositions, marine environment or atmospheric meteorological conditions in the North Atlantic and Northwest Pacific Oceans are different. These different environmental conditions result in different quantitative relations between seawater *Chl-a* and MPOC in these oceanic regions.

Based on the weak correlations between MPOC and EC in marine aerosols (lines 162−164) and the air mass back trajectories (added in Figure S1 and lines 164−166), the influence of transported terrestrial aerosols was limited during the cruises. What's more, the organic aerosols transported from terrestrial environments were secondary or aged organic aerosols, and tend to be watersoluble organic compounds. Thus, the potential terrestrial aerosols contributed the WSOC and MSOC fractions in the marine aerosols.

The seawater *Chl-a* in the parametrization was determined using the spatial average of the satellite-derived *Chl-a* concentrations in Gantt et al. (2011), which was in-situ measured using the collected surface seawater in this work. This could be an additional reason for different parameterizations (added in lines 340−343). Previous studies (Gantt et al., 2011; Vignati et al., 2010) and this study compared the MPOC with WIOC, which is widely regarded from marine primary emissions. Thus, the difference between MPOC and WIOC might not be the main reason for different parameterizations.

**Lines 338−340:**

This is mainly due to different seawater compositions, marine environment or atmospheric meteorological conditions in the North Atlantic and the West Pacific Oceans, which result in different quantitative relations between seawater *Chl-a* and MPOC in these oceanic regions.

**Lines 340−343:**

What's more, the seawater *Chl-a* was determined using the spatial average of the satellite-derived *Chl-a* concentrations in Gantt et al. (2011). This could be an additional reason for the different parameterizations between *Chl-a* and MPOC compared with the results based on the in-suit measured *Chl-a* in this work.

**Lines 162−164:**

The correlation coefficients between OC and EC were lower (Cruise I: r=0.48; Cruise II: r = 0.17) than those between OC and seawater *Chl-a*, suggesting that the potential impacts of transported anthropogenic pollutants were limited during the cruises.

**Lines 164−166:**

The air masses were mainly transported from open oceanic regions, and thus the impacts of terrestrial outflows were limited during the cruises (Fig. S1).

**Newly added Figure S1:**

[Figure]

[Figure]

7.  *For 3.3, overall, I don't see clear conclusion between the two methods, [chl-a] vs [chl-a] + [Na+]. From the "4 Summary", it seems that the authors prefer the second method. However, based on the comparison of Fig. 3b vs 3f and 3c vs 3g, it is not clear to me that the second method performs better. The "r" value is quite the same (0.87 vs 0.88 and 0.73 vs 0.73). Even, the first method performs better by comparing the fraction of points falling between the 1:2 and 2:1 lines in 3b and 3f, which are 69% (L225) and 58% (L243), respectively.*

**Response**: We used both the correlation coefficients (r) and the slopes of the fitting line between WIOC and estimated MPOC to evaluate the performance of different MPOC parameterization approaches. It means that the estimated MPOC shows a similar variation trend to WIOC if with a r value closer to 1, and a good comparison with the WIOC concentrations if with a slope closer to 1. The estimated MPOC using the two methods showed good correlations with WIOC, with the correlation coefficients of 0.87 and 0.88. However, the MPOC concentration was underestimated using [*Chl-a*], with a fitting line slop of 1.17 compared with 1.036 using [*Chl-a*] $\times[Na^+]^{0.45}$ as the input. Related descriptions have been added in lines 302−305, 309−311.

**Lines 302−305:**

Both the correlation coefficients (r) and the slopes of the fitting line between WIOC and estimated MPOC are used to evaluate the performance of different MPOC parameterization approaches. It means that the estimated MPOC shows a similar variation trend to WIOC if with a r value closer to 1, and a good comparison with the WIOC mass concentrations if with a fitting line slope closer to 1.

**Lines 309−311:**

When using p=0.45, both the fitting line slope (1.036) and r value (0.88) suggested an overall better performance than using other *p* values (Fig. S5, 3g). Without the [$Na^+$] as an input parameter, the fitting line slope and r of WIOC and MPOC were respectively 1.17 and 0.87 (Fig. 3b), suggesting an underestimation of MPOC.

8.  *4 and the associated discussion of spatial distribution. How does the season difference affect the spatial distribution?*

**Response**: Aerosol samples were collected among the oceanic region within 15°N−30°N during the two cruises, which were compared to elaborate the seasonal difference. Related analysis has been added in lines 389−396 and Figures S7, S8.

**Lines 386−396:**

Aerosol samples were collected among 15°N−30°N during both the spring and the summer cruises, which were compared to elaborate the seasonal difference. The variations of the estimated MPOC and MSOC along the latitude are shown in Fig. S7, S8. Among the observation region within 15°N−30°N, the average MPOC was comparable in spring (0.16 µgC m$^{-3}$) and summer (0.18 µgC m$^{-3}$), with the average *Chl-a* concentration 0.042 and 0.044 µg L$^{-1}$, respectively. Among the oceanic regions with similar concentrations of seawater *Chl-a*, the MPOC abundance in marine aerosols was comparable without seasonal difference. Among 15°N−30°N, the elevation of MPOC concentrations was consistent with the elevated seawater *Chl-a* concentration without seasonal difference (Fig. S7a). This is consistent with the finding that marine biogenic activities drive the MPOC production. The average MSOC concentration was 0.24 µgC m$^{-3}$ within 15°N−30°N in summer, higher than that in spring (0.19 µgC m$^{-3}$). The elevated MSOC was driven by the increase of seawater TOC concentrations (Fig. S8b). What's more, the stronger solar radiation in summer (Fig. S6) favored the photochemical VOC production in SML, their further photo-oxidation reactions, and the MSOC formation in the atmosphere.

**Newly added Figures S7, S8:**

[Figure]

Figure S7 Spatial distribution of the estimated MPOC and MSOC concentrations. The data is colored by the corresponding seawater *Chl-a* concentrations.

[Figure]

Figure S8 Spatial distribution of the estimated MPOC and MSOC concentrations. The data is colored by the corresponding seawater TOC concentrations.

*9. L281-284: the link between TOC in the seawater and "abiotic VOC precursors from the photochemical production in the sea surface microlayer" is missing. Please further elaborate.*

**Response**: The abiotic VOCs (e.g., isoprene) could be formed via photochemical degradation of soluble organic substances in the surface seawater. We have added their link in lines 376−380.

**Lines 376−380:**

Previous studies suggested that interfacial photochemical degradation of dissolved organic matters in seawater could be an important source of marine VOCs (e.g., isoprene) on a global scale (Bruggemann et al., 2018; Cui et al., 2023; Wang et al., 2023a; Yu and Li, 2021). For remote oceanic regions with high solar radiation but low biological activities, interfacial photochemistry of surface organics could be a major source of abiotic VOCs in the marine boundary layer (Bruggemann et al., 2018; Cui et al., 2023).

*10. L310 "This was consistent to the higher mass contribution by MPOC than MSOC in the marine aerosol samples," There are many samples especially in summer (> half) that have quite low MPOC/OC fractions (< 40%) while the C1 fractions (~ 70% by average) are still very high as the other samples with high MPOC/OC fractions (~ 80%). Therefore, I couldn't buy it with your statement at the current stage. Please further elaborate.*
*And by the way, Fig. S4 doesn't show any measurements of Chl-a (L316)*

**Response**: The difference in MPOC/OC fractions and PRLIS (C1) fractions was due to the different florescent efficiency of organic molecules from different sources. We now have deleted this sentence and added related statements in lines 420−432.

The seawater *Chl-a* data is added in Figure S9.

**Lines 420−432:**

It is noted that the proportion of different fluorescent compounds did not represent their mass contributions, as the florescent efficiency of organic compounds was related to their chemical structures. Organic molecules with substantial conjugation of π-bonds or double bound structures are known to be especially efficient at emitting fluorescence, particularly when N atoms are present (Chen et al., 2016a; Pöhlker et al., 2012). Amino acids, vitamins, and humic-like substances have been identified as efficient fluorophores (Graber and Rudich, 2006; Laskin et al., 2015; Pöhlker et al., 2012). The sea-to-air transfer of phytoplankton-produced protein-containing organics leads to a significant enhancement of fluorescent compounds in SSA (Aller et al., 2017; Lawler et al., 2020; Miyazaki et al., 2018a). The PRLIS, or named protein-like organic matter (PLOM), has been identified as a common component in the oceanic organic matter, and enriched in marine aerosols (Chen et al., 2016b). However, the biogenic SOA (e.g., isoprene oxidation products abundant in marine atmospheres) molecules, without conjugated double bounds, are weakly fluorescent or do not display fluorescent properties (Carlton et al., 2009; Laskin et al., 2015). Thus, the WSOC contributed by biogenic SOA was not included in the detected fluorescent components, and the observed proportions of PRLIS emitted by sea sprays was higher than those of the WIOC mass contribution in the marine aerosols.

**Revised Figure S9:**

[Figure]

Figure S9 Variations of fluorescence component intensity identified by three-component solutions based on PARAFAC model analysis and related aerosol components: (a) C1, Na$^+$, and *Chl-a*, (b) C2 and EC, (c) C3 and SO$_4^{2-}$.

**Minor comments:**

1. *L49 "Ocean surface is one of the largest active reservoirs of organic carbon on Earth, resulting from phytoplankton, algal as well as the related senescence and lysis (Hedges, 1992; Quinn and Bates, 2011)." How large is this reservoir compared to other reservoirs?*

   **Response**: Ocean surface reserve about 18% of the active organic carbon. Other reservoirs include soil humus, land plant tissue and surface marine sediments. We have added the proportion in line 51.

   **Lines 50−52:**

   Ocean surface is one of the largest active reservoirs of organic carbon on Earth (~18%), resulting from phytoplankton, algal as well as the related senescence and lysis (Hedges, 1992; Quinn and Bates, 2011).

2. *L86: please add the details of the source for the satellite-derived chl-a data.*

   **Response**: The details of the satellite-derived *Chl-a* data are now added in the revised version (lines 89−91).

   **Lines 89−91:**

   The satellite-derived *Chl-a* data were provided by Copernicus Marine Environmental Monitoring Service (CMEMS) with a spatial resolution of 4 km and a monthly temporal resolution (https://marine.copernicus.eu/). Here, we utilized the satellite-derived *Chl-a* data during March and June 2022 to support our conclusion.

3. *L97: "organic aerosol concentration". Please be specific. I guess it is "organic aerosol mass concentration".*

   **Response**: Thanks for the reminding. Yes, it is organic aerosol mass concentration. We have revised to be specific (lines 105−106).

   **Lines 105−106:**

   The mass concentration of organic aerosols was calculated by multiplying OC by a conversion factor 1.6 (Wang et al., 2023b).

4. *L116-117: please show their values (at least a selection of the values) to give the audience a direct sense of "comparable".*

   **Response**: Revised accordingly. We now have added the values in lines 132−135.

   **Lines 132−135:**

   The observed OC concentrations during our cruises were comparable to previous studies over the North Pacific Ocean (0.5−0.7 μgC m$^{-3}$), and lower than those observed at an island in the West Pacific Ocean (1.7±1.0 μgC m$^{-3}$) (Hoque et al., 2015; Hoque et al., 2017; Kunwar and Kawamura, 2014).

5. *L138-139: have you tested the difference of the Chl-a concentration between spring and summer? Are they statistically different?*

**Response**: Thanks for the reminding. We test the difference of the *Chl-a* concentrations in spring and summer, and added in lines 168−169.

**Lines 168−169:**

However, the difference was not significant, with a *P* value of 0.33.

*6. In section 3.2, please further explain why we would expect good or poor correlations between chl-a concentration and organic fraction?*

**Response**: As suggested, we have added related explanations in section 3.2 (lines 201−208).

**Lines 201−208:**

Seawater *Chl-a* concentration is one of the most important factors driving the variation of organic fraction in the SSA, and they display good correlations when the wind speed does not vary a lot. However, wind speed should be combined with surface *Chl-a* to predict the organic fraction in SSA if the wind speed varies obviously during the observation or simulation periods (Gantt et al., 2011; Grythe et al., 2014). This is due to the influence of wind on the coverage of sea surface microlayer (SML) in the sea surface, which is enriched in organic compounds. For a given chemical composition of seawater, the largest coverage of sea surface by SML and a higher organic fraction in SSA are expected during calm winds. However, the SML would be destructed by mixing into the underlying seawater and the organic fraction in SSA decreased when surface wind exceeded 8 m s$^{-1}$ (Gantt et al., 2011).

*7. L185 "Under certain marine environment conditions (e.g., Chl-a, wind speed, SST etc.), the abundance of MPOC should be constant." Please further explain what these conditions are with respect to Chl-a, wind speed, SST, etc.*

**Response**: Related descriptions are added in lines 240−245.

**Lines 240−245:**

Seawater *Chl-a* concentration is the most important factors driving the variation of organic fraction in the SSA, and has been widely used to estimate the organic fraction in SSA (Gantt et al., 2011; Vignati et al., 2010). For given chemical composition of seawater, the largest organic fraction in SSA is expected during calm winds. An increase in wind speed above 3–4 m s$^{-1}$ will cause a rapid decrease of organic fraction due to the destructing of the SML coverage, and the lowest organic fraction is expected for wind exceeded 8 m s$^{-1}$ (Gantt et al., 2011). Seawater temperature is related to the production efficiency and the number concentrations of SSA (Christiansen et al., 2019).

*8. Equation L2 and L3: I highly suggest explicitly showing the equation parts like [OC], [Chl-a], etc. in the equations, rather than using those dots. This will make it much easier for readers.*

**Response**: Revised accordingly.

*9.  Fig 3 caption and the texts referring to Fig 3: Please check the reference carefully. For example:*

      1.  the "panels (b, d)" in L208 should be "panels (b, f)".
      2.  The "panels (c, f)" in L209 should be "panels (c, g)".
      3.  L231, 233: Fig. 3d should be "3e"
      4.  L244: 3d should be "3f"

**Response**: Thanks very much for the reminding. We have checked carefully and corrected the reference in Figure 3 caption and related discussion in the main text.

*10.  L242: why "p=0.45"? Why not other values within 0.35-0.65?*

**Response**: We selected p=0.45 based on both the fitting line slope and correlation coefficient of WIOC and estimated MPOC. When using $p=0.45$, both the fitting line slope (1.036) and r value (0.88) suggested an overall better performance than using other $p$ values (Fig. S5). We added the descriptions in lines 309−310 and revised Figure S5 to be clear.

**Lines 309−310:**

When using p=0.45, both the fitting line slope (1.036) and r value (0.88) suggested an overall better performance than using other $p$ values (Fig. S5, 3g).

**Revised Figure S5:**

[Figure]

Figure S5 The variations of the fitting line slopes and correlation coefficients (r) of WIOC and estimated MPOC, using Eq. 3 with the $p$ value changing from 0−1.

*11.  S2 caption: please also add the meaning of the colors.*

**Response**: Revised accordingly.

**Revised Figure S3 caption:**

[Figure]

Figure S3 The variation of $Na^+$ concentration and $Na^+/PM_{2.5}$ as a function of the wind speed during the cruises. The data obtained during the spring Cruise I is in blue, and the data during the summer Cruise II is in red.

**Referee #2**

**General Comment**

*This paper is based on the analysis of PM2.5 samples collected over the Northwest Pacific Ocean during two campaigns in the spring season during 19 Feb.−9 April, 2022, and in the summer during 19 June−30 July, 2022. The chemical composition data have been investigated to the spatial/regional variability but mainly of Marine organic aerosol (MOA) fractions (WSOC and WIOC, primary and secondary), and Chl-a data as a supporting parameter. While the paper presents a quality data set, the discussion of the results is not up to mark. The basis or tracer data is not enough to quantify the contributions of WSOC and WIOC, primary and secondary to MOA.*

*Another concern is the ignoring/undermining of the impact of terrestrial biogenic outflow on the OC levels and their composition. Though the impact of long-range transport is mentioned, evidence (like back trajectory analysis) is not provided.*

*The data have been further utilised to parameterise the marine sources. But it is not well presented or summarized how seasonality in sea surface parameters and prevailing environmental conditions could impact the seasonal MOA composition. Further, following specific comments provide the issues to be addressed.*

> **Response**: Thanks very much for the suggestions and specific comments. We now have carefully addressed the comments and revised the manuscript accordingly.

*(1) Quantifying the contributions of WSOC and WIOC, primary and secondary OC to MOA*

> **Response**: In this work, the contribution of WSOC and WIOC was quantified by the measurement result. The organic carbon (OC) and WSOC in the MOA were quantified by analyzing the collected marine aerosol samples using a Sunset Laboratory thermal/optical carbon analyzer and a TOC analyzer (TOC-L, Shimadzu, Japan). The WIOC in MOA was calculated by the difference between OC and WSOC concentrations in each sample. The contribution of WSOC and WIOC in MOA could then be quantified by the measured concentrations.
>
> We quantified the concentrations and contribution of primary and secondary OC in MOA using the tracers of seawater *Chl-a* and sea salt (Na$^+$). Seawater *Chl-a* or a combination of *Chl-a* and wind is a widely used oceanic tracer to predict the organic fraction in primary marine aerosols (sea spray aerosols) (O'dowd et al., 2004; O'dowd et al., 2008; Rinaldi et al., 2013; Spracklen et al., 2008). Here, we checked the variations of OC or WIOC in MOA as a function of *Chl-a*, wind speed, and sea salt (Na$^+$) in section 3.2, and then proposed a parameterization scheme to estimate the primary and secondary OC in MOA in section 3.3. The performance of MPOC parameterization was then evaluated by comparing the estimated MPOC concentrations with WIOC, which is generally considered as a proxy for MPOC in previous studies.

The estimated primary OC and secondary OC matched well with the WIOC and WSOC, and the performance of the parameterization was better than the widely used functions proposed based on the results in the North Atlantic Ocean (Gantt et al., 2011; Vignati et al., 2010). Thus, we concluded that seawater *Chl-a* and aerosol components (OC, $Na^+$) could be used to separate the primary and secondary OC in MOA. This is an easy approach not dependent on other organic tracers, usually obtained through complex analysis procedures, and this approach is not limited to the time resolution of sample collection.

*(2) The potential impact of terrestrial biogenic outflow on the OC levels and composition*

**Response**: We agree with the reviewer that we cannot exclude the potential impacts of terrestrial biogenic outflows on the OC levels or compositions in the MOA. The OC contributed by terrestrial outflows is aged/secondary organic aerosols, which are mostly included in the WSOC and the oxygenated humic-like substances (HULIS, identified C3 in the fluorescence components) in MOA. Previous studies have suggested that aged or secondarily formed organic aerosols are mostly water-soluble (Boreddy et al., 2018; De Jonge et al., 2024; Miyazaki et al., 2010), as described in lines 187−189. The impacts of terrestrial biogenic outflows on OC compositions were included in and could be partially reflected by the variation of oxygenated HULIS (C3) in the collected aerosol samples. C3 showed a similar variation to sulfate, which indicated their sources related to secondary formation or aged processes. Oxygenated HULIS included the secondarily-formed and aged organic aerosols from both marine and terrestrial sources. Related analysis is now added in lines 318−322, 411−412.

As suggested, the back trajectory analysis of air masses during the cruises is now added in the revised version (Figure S1). Based on the back trajectories of air masses and the weak correlation between OC and EC (a tracer of terrestrial combustion emissions), we proposed that the impact of terrestrial outflows might be limited during the cruises. The air masses were mainly transported from open ocean based on the back trajectory analysis. What's more, the correlation coefficients between OC and EC (Cruise I: r=0.48; Cruise II: r = 0.17) were lower than those between OC and seawater *Chl-a* (Cruise I: r = 0.81, p < 0.01; Cruise II: r = 0.67, p < 0.01), as stated in lines 162−164. This suggested that the potential impacts of transported terrestrial outflows were limited compared to the marine biological emissions during the cruises. Related analysis is now added in lines 95−96, 322−324, 369−370.

**Lines 318−322:**

It is noted that, based on the shipboard in-situ observation, we cannot exclude the potential impacts of gaseous precursors or aged organic aerosols long-range transported from terrestrial environments, which were mostly in the MSOC fraction. The organic aerosols transported from terrestrial environments were secondary or aged organic aerosols, and tend to be water-soluble organic compounds (Boreddy et al., 2018; De Jonge et al., 2024; Miyazaki et al., 2010).

**Lines 411−412:**

Oxygenated HULIS included the secondarily-formed and aged organic aerosols from both terrestrial and marine sources.

**Lines 95−96:**

The 24-hr backward trajectories of air masses (Fig. S1) originating at 500 m above the ground level were calculated along the observation cruises every 24 hr using the HYSPLIT model (Version 5.2.1, NOAA).

**Lines 322−324:**

Based on the air mass back trajectories (Fig. S1) and the weak correlations between OC and EC stated in section 3.1, the impacts of transported continental outflows were limited during the cruises.

**Lines 369−370:**

Based on the air mass back trajectories (Fig. S1), the impacts of transported terrestrial outflows were limited among the observation regions.

**Newly added Figure S1:**

[Figure]

[Figure]

Figure S1 The 24-hr back trajectories of air masses during the cruises in (a) spring and (b) summer.

*(3) Seasonal variations of sea surface parameters and MOA compositions*

**Response**: Aerosol samples were collected among the oceanic region within 15°N−30°N during the two cruises. The sea surface parameters (*Chl-a*, TOC) and MOA compositions within 15°N−30°N were compared to elaborate the seasonal difference. Related analysis has been added in lines 386−396 and Figures S7, S8.

Aerosol samples were collected among 15°N−30°N during both the spring and the summer cruises, which were compared to elaborate the seasonal difference. The variations of the estimated MPOC and MSOC along the latitude are shown in Fig. S7, S8. Among the observation region within 15°N−30°N, the average MPOC was comparable in spring (0.16 μgC m$^{-3}$) and summer (0.18 μgC m$^{-3}$), with the average *Chl-a* concentration 0.042 and 0.044 μg L$^{-1}$, respectively. Among the oceanic regions with similar concentrations of seawater *Chl-a*, the MPOC abundance in marine aerosols was comparable without seasonal difference. Among 15°N−30°N, the elevation of MPOC concentrations was consistent with the elevated seawater *Chl-a* concentration without seasonal difference (Fig. S7a). This is consistent with the finding that marine biogenic activities drive the MPOC production. The average MSOC concentration was 0.24 μgC m$^{-3}$ within 15°N−30°N in summer, higher than that in spring (0.19 μgC m$^{-3}$). The elevated MSOC was driven by the increase of seawater TOC concentrations (Fig. S8b). What's more, the stronger solar radiation in summer (Fig. S6) favored the photochemical VOC production in SML, their further photo-oxidation reactions, and the MSOC formation in the atmosphere.

**Newly added Figures S7, S8:**

[Figure]

Figure S7 Spatial distribution of the estimated MPOC and MSOC concentrations. The data is colored by the corresponding seawater *Chl-a* concentrations.

[Figure]

Figure S8 Spatial distribution of the estimated MPOC and MSOC concentrations. The data is colored by the corresponding seawater TOC concentrations.

**General Comment**

*Line 57-58: "regions with high emission rates of MOA are largely related to the spatial" Why high emissions rates only? You may revise "regional emission rates of MOA are largely related to the spatial"*

    **Response**: Revised accordingly (line 58).

    **Lines 58−59:**

    A recent modeling study suggested that regional emission rates of MOA are largely related to the spatial distribution of ocean biological productivity (Zhao et al., 2021).

*Line 60: "Due to the relationship between sea surface phytoplankton and MOA" I feel this phrase is not required.*

    **Response**: Deleted as suggested.

*Line 64: "Observation-based parameterization..." Provide a reference to support this sentence.*

    **Response**: Related references (Brooks and Thornton, 2018b; Quinn et al., 2015a) are now cited.

*Line 77: As shown in Figure 1(a) the track covers between 02S-34 N, so it would be appropriate to mention the region as "Northwest Pacific Ocean"? covering southern latitudes*

    **Response**: Thanks for the reminding. We change "the Northwest Pacific Ocean" to "the West Pacific Ocean (WPO)" throughout the main text in the revised version.

*Line 80-81: "suspended particles (TSP) samples and PM2.5" can be revised as "suspended particles (TSP) and PM2.5"*

    **Response**: Revised accordingly.

*Line 114: "observation, and" should be "observation and"*

**Response**: Revised accordingly.

*Figure 1(b). I suggest to mention the year of the campaigns. You may provide a common x-axis title as (Date/Year).*

**Response**: Thanks for the suggestion. The x-axis title of Figure 1 has been revised accordingly, and the year of the campaigns is mentioned.

**Revised Figure 1:**

[Figure]

*Line 131: "r = 0.67, p < 0.01, Fig. 1)," r= 0.67 may not represent a strong correlation. Also I dont find correlation plots in Figure 1, perhaps referring to Fig 2?*

**Response**: Thanks for the reminding. This sentence is revised accordingly (lines 158−159).

**Lines 158−159:**

The abundance of MOA displayed similar spatial distribution (Fig. 1) and strong or medium correlations with the sea surface *Chl-a* concentration…

*Line 136: But how about the impact/outflow of terrestrial biogenic emissions. ?*

**Response**: As suggested in the general comment, we now have added the back trajectory analysis of air masses to address the potential impacts of terrestrial biogenic emissions during the cruises. The air masses were mainly transported from open oceanic regions based on the back trajectory analysis. The potential impacts of transported terrestrial outflows could be limited during the cruises. Related analysis is added in lines 95−96, 164−166 and Figure S1.

We agree with the reviewer that we cannot exclude the potential impacts of terrestrial biogenic outflows on OC in the marine aerosol samples. Based on the data in this work, we cannot distinguish the contribution of terrestrial biogenic emissions from the WSOC or the oxygenated HULIS. We will try to evaluate this impact combining other organic tracers in our future studies.

**Lines 95−96:**

The 24-hr backward trajectories of air masses (Fig. S1) originating at 500 m above the ground level were calculated along the observation cruises every 24 hr using the HYSPLIT model (Version 5.2.1, NOAA).

**Lines 164−166:**

The air masses were mainly transported from open oceanic regions, and thus the impacts of terrestrial outflows were limited during the cruises (Fig. S1).

**Newly added Figure S1:**

[Figure]

Figure S1 The 24-hr back trajectories of air masses during the cruises in (a) spring and (b) summer.

*Line 138-139: "spring 138 (0.09 ± 0.06 µg L-1) than in summer (0.07 ± 0.05 µg L-1)." I think the values are not very different. So the explanation for the differences in OC levels is not robust. I feel the transport from regions of Papua New Guinea and Indonesia needs a consideration. Following works may provide more insights.*
*https://doi.org/10.1038/s41612-022-00311-0*
*https://doi.org/10.5194/essd-15-5403-2023*

**Response**: We checked the significant level of the difference, and stated that the difference is not significant (*P*=0.33).

Thanks for your reminding of the related literatures. Based on the back trajectories of air masses, the influence of transported terrestrial emissions was limited. The potential influence of transported MOA or biogenic VOC precursors from the coastal oceanic regions of Papua New Guinea and Indonesia is now described in lines 170−173.

**Lines 170−173:**

For the samples collected near the equator, the MOA or biogenic VOC precursors could also be transported from coastal oceanic regions of Papua New Guinea and Indonesia with higher marine biological activity and higher isoprene emission fluxes (Cui et al., 2023; Zhang and Gu, 2022). This could be an additional reason for the higher OC level during the spring cruise and the highest OC concentration observed on 14 March.

*Line 141-142 and other places: "mass ratios of 70% ± 27% in spring and 48% ± 35% in" should be "mass ratios of 70 ± 27% in spring and 48 ± 35% in"*

**Response**: Revised accordingly.

*Line 145: "WSPC in marine aerosols." should be "WSOC in marine aerosols."*

**Response**: It is corrected.

*Line 147-148: "and transferred to atmospheric aerosols over the ocean." Rewrite with a better phrase.*

**Response**: This sentence is rewritten to be clear (lines 184−185).

**Lines 184−185:**

These organic substances could be enriched in the surface seawater and then transferred into the atmospheric aerosols within the marine boundary layer.

*Line 152: "marine reactive organic gases (Boreddy et al., 2018; De Jonge et al., 2024; Miyazaki et al., 2010)." Here I would suggest citing studies which provide measurements of reactive trace gases implying the roles in SOA in marine air. Following works may be cited:*
*Sources and distribution of light NMHCs in the marine boundary layer of the northern Indian Ocean during winter: Implications to aerosol formation. Journal of Geophysical Research: Atmospheres, 129, e2023JD039433. https://doi.org/10.1029/2023JD039433*
*Elevated levels of biogenic nonmethane hydrocarbons in the marine boundary layer of the Arabian Sea during the intermonsoon. Journal of Geophysical Research: Atmospheres, 125, e2020JD032869. https://doi.org/10.1029/2020JD032869..*

**Response**: Thanks for your kind reminding of the related papers. We now have cited these works in the revised manuscript (lines 189−191).

**Lines 189−191:**

Reactive gaseous precursors of organic aerosols are widely observed over different oceanic regions (Tripathi et al., 2024; Tripathi et al., 2020; Wang et al., 2023a), which contribute to the SOA formation in the marine boundary layer.

*Further, in this section, I suggest comparing the composition of WSOC, WISOC, and other species presented in typically continentally influenced air over this region. The*

*following work may be useful in this regard (It would be good if you can find some other similar works and highlight those)*

*Anthropogenic aerosols observed in Asian continental outflow at Jeju Island, Korea, in spring 2005, J. Geophys. Res., 114, D03301, doi:10.1029/2008JD010306.*

**Response**: We have cited related works and compared the concentrations or contributions of WSOC, WIOC and EC in this work and those observed in the atmospheres typically influenced by continental outflows (lines 130−132, 174−179).

**Lines 130−132:**

The EC concentrations were $0.066 \pm 0.056$ μgC m$^{-3}$ and $0.055 \pm 0.052$ μgC m$^{-3}$ during the spring and the summer observations, much lower than those observed over coastal areas typically influenced by continental outflows (Sahu et al., 2009; Zhang et al., 2025).

**Lines 174−179:**

The observed concentrations of WSOC and WIOC over the open Pacific Ocean were lower than those observed in the atmosphere under severe influence of continental outflows (Sahu et al., 2009; Zhang et al., 2025). Marine organic aerosols were dominated by the water-insoluble fractions, with the WIOC/OC mass ratios of $70 \pm 27\%$ in spring and $48 \pm 35\%$ in summer (Fig. 1). The proportion of water-soluble organics in MOA over the WPO was lower than that observed over the East Asian marginal seas in autumn (75%), during which severe impacts of continental anthropogenic pollutants were observed (Zhang et al., 2025).

*Line 155: "..through the ocean bubble bursting." why through bubble bursting only as other air-sea exchange processes can also contribute?. Please explain and revise accordingly.*

**Response**: The seawater organic substances could be transferred into sea spray aerosols through bubble bursting and wave breaking processes. We now have revised this sentence accordingly (lines 193−194).

**Lines 193−194:**

The similar variation trends and good correlations between WIOC in marine aerosols and seawater *Chl-a* (Fig. 1, 2) suggested the origins of MOA from seawater through ocean bubble bursting or wave breaking.

*Line 158-160: "in the present .." I think similar discussion is presented in previous section (line 143-144). I suggest to concise the discussion related to correlation in one section only.*

**Response**: This sentence is simplified accordingly (lines 197−198). We try to address the difference between WIOC and WSOC in section 3.1 (lines 179−181 in the revised version), and the difference between organic concentration and organic fractions in section 3.2 (lines 197−198 in the revised version).

Over the West Pacific Ocean, we observed better correlations between OC or WIOC concentrations and *Chl-a* than those between organic or water-insoluble organic mass fractions and *Chl-a* (Fig. 2a−2d).

*Line 170: "concentrations of OC or WIOC in PM2.5 did not show obvious correlations with" I disagree here. The levels OC and WIOC tend to decrease rapidly with wind speed (may not be linear). The discussion should be based on dependencies or trend (with Chl-a and wind speed) but not strictly in terms of correlation/ anti-correlation (as data points are scattered)*

**Response**: Thanks for the reminding. This sentence is revised accordingly (lines 216−217).

**Lines 216−217:**

During our cruises over the WPO, the concentrations of OC or WIOC in PM$_{2.5}$ showed a decreasing trend with the increase of wind speed (Fig. 2e, 2f).

*Line 172: "which was due to the elevated proportions of inorganic sea salts in the marine aerosols under the high-wind" This justification is not convincing. That can change only the magnitude of the slope but not the direction (positive/negative), check.*

**Response**: The reason is revised as (lines 218−219): The organic-enriched SML in the sea surface would be destructed under high wind speed conditions, which results in a decrease of organic substances transported into the SSA (Gantt et al., 2011).

*Line 185: "Under certain marine environment conditions (e.g., Chl-a, wind speed, SST etc.)," What is certain here , as values of these parameters are not given. And why MPOC should be constant anyway?*

**Response**: Surface seawater *Chl-a* and wind speed are the most important marine environmental factors determining the abundance of primary marine organic aerosols (organics in sea spray aerosols). Thus, for a given *Chl-a* and wind speed condition, the abundance of MPOC primarily generated from surface seawater should be constant. We now have added related descriptions to be clear (lines 238−245).

**Lines 238−245:**

For a given marine environment condition (a given *Chl-a*, wind speed, sea surface temperature (SST), etc.), the abundance of MPOC should be constant (Gantt et al., 2011). Seawater *Chl-a* concentration is the most important factors driving the variation of organic fraction in the SSA, and has been widely used to estimate the organic fraction in SSA (Gantt et al., 2011; Vignati et al., 2010). For given chemical composition of seawater, the largest organic fraction in SSA is expected during calm winds. An increase in wind speed above 3–4 m s$^{-1}$ will cause a rapid decrease of organic fraction due to the destructing of the SML coverage, and the lowest organic fraction is expected for wind exceeded 8 m s$^{-1}$ (Gantt et al., 2011). Seawater temperature is related to the production efficiency and the number concentrations of SSA (Christiansen et al., 2019).

*Line 188: "Based on the shipboard observations in present study," Should be "In the present study,"*

**Response**: Revised accordingly.

*Line 191: "contribution of SOA (e.g., methanesulfonic acid from DMS oxidation," Check if DMS oxidation provides SOA, I know it gives sulfate. But then do you consider sulfate as SOA ? Check following and other papers on DMS*
*Processes controlling DMS variability in marine boundary layer of the Arabian Sea during post-monsoon season of 2021. Journal of Geophysical Research: Atmospheres, 130, e2024JD042547. https://doi.org/10.1029/2024JD042547*

**Response**: The oxidation of DMS could form both sulfate and methanesulfonic acid (MSA) (Barnes et al., 2006). MSA is an important SOA compound in the marine boundary layer. Related papers are cited in the main text (lines 251−252).

*Line213-214: "Based on the correlation analysis above," should be "Based on the correlation analysis"*

**Response**: Revised accordingly.

*Line 215-219: "In the classic OC/EC ratio method...Yu et al., 2021)." These sentences appear out of context , may be removed.*

**Response**: We have deleted these sentences here as suggested.

*Figure 4 (a-d): Y-axis should be Latitude . Also check Fig 4 (e-f) if axes titles are correct (latitude /longitude or vice versa) .*

**Response**: Thanks for the reminding. We now have corrected the axes titles of Figure 4.

**Revised Figure 4:**

[Figure]

*Line 272:"high concentrations over the oceanic regions among 5°S–5°N and 35°N–40°N," Here for these two region, why the contributions from the land vegetation should also be discussed (due to vicinity to land/island regions)*

**Response**: The potential contributions of organic aerosols from land or coastal regions are analyzed in lines 369−373.

**Lines 369−373:**

Based on the air mass back trajectories (Fig. S1), the impacts of transported terrestrial outflows were limited among the observation regions. Marine organic aerosols or biogenic VOC precursors could also be transported from coastal oceanic regions with higher *Chl-a* levels and higher isoprene emission fluxes (Cui et al., 2023; Zhang and Gu, 2022), which could be an additional reason for the higher MOA concentrations within 5°S–5°N and 35°N–40°N.

*LIne 279-280" "from surface seawater as well as the production of biologic VOCs (e.g., isoprene, DMS, etc.) from phytoplankton" should be revised as "from seawater as well as the production of VOCs from phytoplankton" but again terrestrial contributions should be discussed.*

**Response**: Revised accordingly.

*Line 284-286: "The strong.." This sentence seems speculative also Chl-a map does*

*not support this localized feature.*

**Response**: Surface net solar radiation (SSR) data is now added to address the stronger solar radiation within 15°N–20°N during the summer cruise than during spring (lines 93−95, 382−383, Figure S6).

**Lines 93−95:**

Surface net solar radiation (SSR) data were obtained from the hourly data of the ECMWF Reanalysis v5 (ERA5) product (Hersbach et al., 2020), with a spatial resolution of 0.25°.

**Lines 382−383:**

The strong solar radiation during the summertime (19 June- 30 July) Cruise II, as shown in Fig. S6, favored the photochemical VOC production and the SOA formation in marine atmospheres.

**Newly added Figure S6:**

[Figure]

Figure S6 Diurnal variation of the surface net solar radiation (SSR) within the 15°N–20°N during the spring and the summer cruises.

**Lines 279−285:**

[revised manuscript text omitted]